# PERTURBED DYNAMIC TIME WARPING: A PROBABILISTIC FRAMEWORK AND GENERALIZED VARIANTS

**Xiangqian Sun**[*1]**, Chaoqun Wang**[2]**, Wei Zhang**[3]

[1]School of Mathematics and Physics, Xi'an Jiaotong-Liverpool University, China
[2]School of AI and Advanced Computing, Xi'an Jiaotong-Liverpool University, China
[3]School of Management, Zhejiang University, China
{Xiangqian.Sun, Chaoqun.Wang}@xjtlu.edu.cn, wei_zhang@zju.edu.cn

## ABSTRACT

Dynamic Time Warping (DTW) is a classical method for measuring similarity between time series, but its non-differentiability hinders integration into end-to-end learning frameworks. To address this, soft-DTW replaces the minimum operator with a smooth soft-min, enabling differentiability and efficient computation. Motivated by soft-DTW, we propose perturbed-DTW, a differentiable framework of DTW obtained by adding random perturbations to warping costs and taking the expected minimum. Under Gumbel noise, perturbed-DTW exactly recovers soft-DTW, providing a natural probabilistic interpretation of soft-DTW. We further generalize this framework by extending the Gumbel noise to the broader family of generalized extreme value (GEV) distributions, leading to a new class of soft-DTW variants. Building on this insight, we introduce nested-soft-DTW (ns-DTW), which integrates GEV perturbations into the dynamic programming formulation of perturbed-DTW. This extension induces alignments with tunable skewness, offering greater flexibility in modeling diverse alignment structures. We validate ns-DTW on barycenter computation, clustering, and classification, demonstrating its effectiveness over existing approaches.

## 1 INTRODUCTION

Dynamic Time Warping (DTW) is a classical measure of similarity between time series that computes the minimum-cost alignment between two sequences (Berndt & Clifford, 1994; Sakoe & Chiba, 2003). Unlike Euclidean distance, DTW accommodates temporal distortions and unequal sequence lengths, making it broadly applicable across domains such as object recognition (Belongie et al., 2002), time-series forecasting (Le Guen & Thome, 2019), and irregular sequence modeling (Zhang et al., 2023). Despite its effectiveness, DTW is limited by the non-differentiable nature of its minimum operator, rendering it incompatible with gradient-based optimization methods.

To address non-differentiability, Cuturi & Blondel (2017) introduced *soft-DTW*, which replaces the hard minimum with a smooth soft-min operator. Soft-DTW admits efficient dynamic programming and yields gradients that can be computed recursively. Thereby, it enables gradient-based optimization in applications such as music score alignment (Mensch & Blondel, 2018), video segmentation (Chang et al., 2019) and trajectory clustering (Brankovic et al., 2020).

Motivated by soft-DTW, we extend the differentiable DTW framework through the lens of *perturbed optimizer* (Berthet et al., 2020). We propose **perturbed-DTW**, a new scheme that introduces randomness into the alignment costs, computes the minimum perturbed cost, and then averages over the noise distribution. This formulation naturally yields a differentiable relaxation: the optimal alignment matrix becomes a distribution over paths rather than a single deterministic solution. Interestingly, when the perturbations are Gumbel distributed, perturbed-DTW recovers soft-DTW. This provides a new probabilistic interpretation of soft-DTW as the expectation of DTW under Gumbel perturbations.

---

[*]corresponding author

Building on perturbed-DTW, we introduce **nested-soft-DTW (ns-DTW)**, a novel variant obtained by employing the generalized extreme value (GEV) distribution as the perturbation. By modeling correlations across groups of variables, ns-DTW captures richer alignment structures. Remarkably, the resulting alignments of ns-DTW can exhibit tunable skewness beyond what soft-DTW allows.

Our contributions are threefold:

- We introduce perturbed-DTW, a general perturbation-based framework for differentiable DTW. Within this framework, soft-DTW emerges naturally as the expectation of DTW when alignment costs are perturbed by Gumbel noise, thereby providing a probabilistic interpretation of its smoothing behavior.
- By adopting GEV perturbations, we derive ns-DTW, which offers greater modeling flexibility through correlated perturbations and skewed alignment distributions.
- We demonstrate the effectiveness of ns-DTW on diverse time-series tasks, showing that it captures meaningful alignment structures while remaining computationally tractable.

The remainder of the paper is organized as follows. Section 2 reviews DTW and soft-DTW. Section 3 presents perturbed-DTW and its connection to soft-DTW, and introduces ns-DTW and its properties. Section 4 provides experimental results across benchmark datasets.

## 2 PRELIMINARIES

### 2.1 DYNAMIC TIME WARPING

Consider two $p$-dimensional time series $\mathbf{x} = [\boldsymbol{x}_1, \ldots, \boldsymbol{x}_m] \in \mathbb{R}^{p \times m}$ and $\mathbf{y} = [\boldsymbol{y}_1, \ldots, \boldsymbol{y}_n] \in \mathbb{R}^{p \times n}$. Denote $[m] = \{1, \ldots, m\}$ and $[n] = \{1, \ldots, n\}$. *Dynamic time warping* (DTW) aims to find the optimal alignment between two sequences by allowing a point in one sequence to be matched with one or more points in the other. To formulate the optimal alignment problem, we first define the alignment cost. A *local cost matrix* $\boldsymbol{C} \in \mathbb{R}^{m \times n}$ measures element-wise dissimilarities: $[\boldsymbol{C}(\mathbf{x}, \mathbf{y})]_{i,j} = c(\boldsymbol{x}_i, \boldsymbol{y}_j)$, where $c(\cdot, \cdot) : \mathbb{R}^p \times \mathbb{R}^p \to \mathbb{R}_+$ is a differentiable cost function. A commonly used cost function adopts the squared Euclidean distance, that is,

$$[\boldsymbol{C}(\mathbf{x}, \mathbf{y})]_{i,j} = c(\boldsymbol{x}_i, \boldsymbol{y}_j) = \frac{1}{2} \|\boldsymbol{x}_i - \boldsymbol{y}_j\|_2^2, \quad \forall i \in [m], \, j \in [n].$$

Then, an *alignment matrix* $\boldsymbol{A} \in \{0, 1\}^{m \times n}$ encodes an alignment between data points $\boldsymbol{x}_i$ and $\boldsymbol{y}_j$: $A_{i,j} = 1$ if $\boldsymbol{x}_i$ is aligned with $\boldsymbol{y}_j$, and $0$ otherwise. In addition, we call a alignment matrix is *valid* if it satisfies: (i) the nonzero entries of $\boldsymbol{A}$ form a path starting from $(1,1)$ and ending at $(m,n)$; (ii) the moves in the path can only be one of the directions: $\{\rightarrow, \downarrow, \searrow\}$. Denote the set of *all valid alignment matrices* as $\mathcal{A}_{m,n}$. Given the local cost matrix $\boldsymbol{C}$ and the alignment matrix $\boldsymbol{A} \in \mathcal{A}_{m,n}$, DTW is defined as

$$\text{DTW}(\boldsymbol{C}) := \min_{\boldsymbol{A} \in \mathcal{A}_{m,n}} \langle \boldsymbol{A}, \boldsymbol{C} \rangle, \tag{1}$$

where $\langle \boldsymbol{A}, \boldsymbol{C} \rangle = \text{Trace}(\boldsymbol{C}^\top \boldsymbol{A})$ is the sum of elementwise products (Frobenius inner product). *Optimal alignment matrix* $\boldsymbol{A}^*$ is defined as the one that achieves the minimum cost

$$\boldsymbol{A}^* = \arg\min_{\boldsymbol{A} \in \mathcal{A}_{m,n}} \langle \boldsymbol{A}, \boldsymbol{C} \rangle. \tag{2}$$

However, directly computing DTW via (1) is intractable due to the exponential size of $\mathcal{A}_{m,n}$. Instead, DTW can be computed efficiently via dynamic programming (DP). To formulate the DP, define $D_{i,j}$ as the minimal cost of alignment from time series $[\boldsymbol{x}_1, \ldots, \boldsymbol{x}_i]$ and $[\boldsymbol{y}_1, \ldots, \boldsymbol{y}_j]$. Then, $D_{i,j}$ satisfies the Bellman equation (Bellman, 1952),

$$D_{i,j} = \min \Big\{ \underbrace{D_{i,j-1}}_{A_{i,j-1}=1}, \underbrace{D_{i-1,j}}_{A_{i-1,j}=1}, \underbrace{D_{i-1,j-1}}_{A_{i-1,j-1}=1} \Big\} + C_{i,j}, \quad \forall 1 < i \leq m, 1 < j \leq n. \tag{3}$$

with boundary conditions $D_{i,1} = \sum_{k=1}^{i} C_{k,1}$, $\forall i \in [m]$, and $D_{1,j} = \sum_{k=1}^{j} C_{1,k}$, $\forall j \in [n]$. Then DTW distance is then given by $D_{m,n}$, i.e., $\text{DTW}(\boldsymbol{C}) = D_{m,n}$. Therefore, the optimal alignment

path can be recovered by backtracking from $D_{m,n}$ to $D_{1,1}$ from (3). This approach guarantees that the optimal solution is found by considering all possible alignment paths while avoiding the exponential complexity of exhaustive enumeration.

Equation (3) enables efficient computation of both the DTW distance and its optimal alignment via dynamic programming. Yet, the inherent non-differentiability of DTW, stemming from the minimum operator, precludes its use in end-to-end learning pipelines. Introducing differentiability without sacrificing computational efficiency is a central challenge in extending DTW to modern learning frameworks (Cuturi & Blondel, 2017).

## 2.2 SOFT-DTW

Cuturi & Blondel (2017) proposed *soft-DTW*, a differentiable relaxation of DTW that replaces the minimum operator in Equation (1) with a smooth approximation (Cuturi et al., 2007; Saigo et al., 2006) . Technically, for a vector $\boldsymbol{x} = (x_1, \ldots, x_n)$, the *soft minimum* operator is defined as

$$\min{}_\gamma \boldsymbol{x} = -\gamma \log \sum_{i=1}^{n} \exp(-\frac{x_i}{\gamma}) \,,$$

where $\gamma > 0$ is the *temperature parameter* to control the smoothness and bias. In particular, $\min_\gamma \boldsymbol{x}$ approaches $\min \boldsymbol{x}$ as $\gamma \to 0$ while it approaches $-\gamma \log n$ as $\gamma \to \infty$. Formally, soft-DTW is defined as

$$\text{soft-DTW}_\gamma(\boldsymbol{C}) := \min{}_\gamma \langle \mathbf{A}, \mathbf{C} \rangle = -\gamma \log \sum_{\boldsymbol{A} \in \mathcal{A}_{m,n}} \exp\left(-\frac{\langle \boldsymbol{A}, \boldsymbol{C} \rangle}{\gamma}\right), \tag{4}$$

where $\langle \mathbf{A}, \mathbf{C} \rangle := \big(\langle \boldsymbol{A}, \boldsymbol{C} \rangle\big)_{\boldsymbol{A} \in \mathcal{A}_{m,n}} \in \mathbb{R}^{|\mathcal{A}_{m,n}|}$ denotes the vector whose entries are the alignment costs $\langle \boldsymbol{A}, \boldsymbol{C} \rangle$ concatenated over all alignments in $\mathcal{A}_{m,n}$. The soft-minimum operator here maps vector of all possible alignment costs into an "aggregated" cost. Noted that soft-DTW is a differentiable discrepancy compared to DTW, and it converges to DTW when $\gamma \to 0$. Conversely, as $\gamma \to \infty$, soft-DTW approaches the mean of all possible alignment costs.

The differentiability of soft-DTW with respect to both the time series $\mathbf{x}$, $\mathbf{y}$ and the cost matrix $\boldsymbol{C}$ enables its use in gradient-based learning algorithms. This property makes it particularly valuable for applications in machine learning and optimization. Notably, the optimal alignment matrix in soft-DTW is no longer deterministic; instead, it follows a Gibbs distribution over $\mathcal{A}_{m,n}$ :

$$P(\boldsymbol{A}; \boldsymbol{C}) = \frac{\exp\big(-\langle \boldsymbol{A}, \boldsymbol{C} \rangle / \gamma\big)}{\sum_{\boldsymbol{A}' \in \mathcal{A}_{m,n}} \exp\big(-\langle \boldsymbol{A}', \boldsymbol{C} \rangle / \gamma\big)}. \tag{5}$$

The expected alignment matrix is

$$\boldsymbol{E} = \sum_{A \in \mathcal{A}_{m,n}} P(\boldsymbol{A}; \boldsymbol{C}) \cdot \boldsymbol{A}.$$

Here, $E_{i,j} \in (0,1)$ represents the marginal probability that $\boldsymbol{x}_i$ is aligned with $\boldsymbol{y}_j$. Although soft-DTW has a succinct form, direct evaluating (4) is computationally challenging, as it requires summing over all possible alignment matrices. However, Mensch & Blondel (2018) showed that soft-DTW can be reformulated as entropy-regularized dynamic programming. Specifically, one can obtain the *soft accumulated cost matrix* $\boldsymbol{S}$ by replacing the hard minimum in the DTW recursion (3) with the soft minimum:

$$S_{i,j} = \min{}_\gamma \big\{ S_{i-1,j-1}, \ S_{i-1,j}, \ S_{i,j-1} \big\} + C_{i,j}.$$

This recursion yields $\text{soft-DTW}_\gamma(\boldsymbol{C}) = S_{m,n}$ (Cuturi & Blondel, 2017). Additionally, soft-DTW admits an alternative variational formulation, expressed as the solution to an entropy-regularized linear program (Blondel et al., 2021). Let

$$\boldsymbol{p}(\boldsymbol{C}) := \big(P(\boldsymbol{A}; \boldsymbol{C})\big)_{\boldsymbol{A} \in \mathcal{A}_{m,n}} \in \Delta^{|\mathcal{A}_{m,n}|} \quad \text{and} \quad \boldsymbol{s}(\boldsymbol{C}) := \langle \mathbf{A}, \mathbf{C} \rangle \in \mathbb{R}^{|\mathcal{A}(m,n)|}.$$

denote the vector of alignment matrix probabilities and its corresponding alignment costs, respectively.[1] Thus, soft-DTW can be expressed in the variational form:

$$\text{soft-DTW}_\gamma = \min_{\boldsymbol{p} \in \Delta^{|\mathcal{A}_{m,n}|}} \langle \boldsymbol{p}, \boldsymbol{s} \rangle + \gamma H(\boldsymbol{p}), \tag{6}$$

where $H(\boldsymbol{p}) = \langle \boldsymbol{p}, \log \boldsymbol{p} \rangle$ denotes the negative Shannon entropy. This variational perspective highlights soft-DTW as an entropy-regularized alignment cost (Blondel et al., 2020; Sun et al., 2023), as illustrated in Figure 1

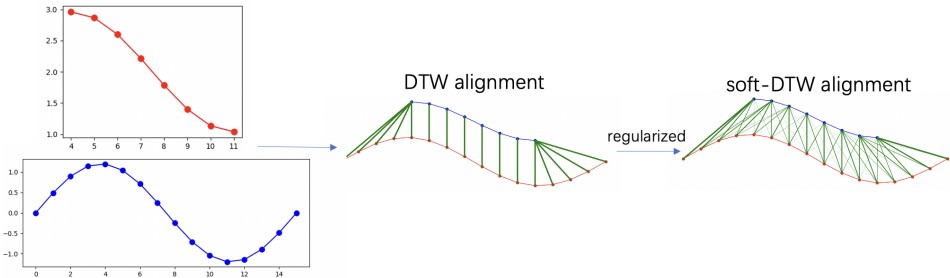

Figure 1: An illustration of soft-DTW. The right one is the expected alignment of soft-DTW.

## 3 PERTURBED DYNAMIC TIME WARPING

### 3.1 PERTURBED-DTW DEFINITION

In this section, we introduce perturbed-DTW, a new scheme that provides an alternative means of rendering the original DTW differentiable. Let $\boldsymbol{x} = (x_1, \ldots, x_n)^\top \in \mathbb{R}^n$ and let $\boldsymbol{\varepsilon} = (\varepsilon_1, \ldots, \varepsilon_n)^\top$ denote a perturbation vector drawn from a distribution $\mathbb{P}$. For temperature parameter $\gamma > 0$, define the perturbed minimum as

$$\mathbb{E}_{\boldsymbol{\varepsilon} \sim \mathbb{P}}\big[ \min\{\boldsymbol{x} - \gamma \boldsymbol{\varepsilon}\} \big] = \mathbb{E}_{\boldsymbol{\varepsilon} \sim \mathbb{P}}\big[ \min\{x_1 - \gamma \varepsilon_1, \cdots, x_n - \gamma \varepsilon_n\} \big]. \tag{7}$$

Next, we define the perturbed-DTW by replacing the standard minimum operator in the DTW with this perturbed version.

**Definition 1.** *The perturbed-DTW is defined as*

$$\text{perturbed-DTW}_\gamma(C) \coloneqq \mathbb{E}_{\boldsymbol{\varepsilon} \sim \mathbb{P}} \left[ \min \left\{ \langle \mathbf{A}, \mathbf{C} \rangle - \gamma \boldsymbol{\varepsilon} \right\} \right] \tag{8}$$

*where $\boldsymbol{\varepsilon}$ is a perturbation vector of dimension $|\mathcal{A}_{m,n}|$ distributed according to $\mathbb{P}$, and $\gamma > 0$ is the temperature parameter.*

Intuitively, our method first perturbs the cost of each valid alignment with a random noise term $\gamma \varepsilon$, and then takes the minimum over all possible alignment matrices. The final result is the expectation of this minimum value with respect to the probability distribution of the noise. The expected alignment under perturbed-DTW can be treated as an aggregated version of DTW alignments under different realizations of $\boldsymbol{\varepsilon}$, as shown in Figure 2 . This formulation is closely related to **random utility theory** (Train, 2009) , where choices are made based on utilities perturbed by random shocks. Unlike soft-DTW, which relies on a heuristic soft-minimum operator, perturbed-DTW achieves differentiability through randomization. This approach establishes a deep connection between time series alignment and random utility models, where an alignment matrix $A$ represents a choice with a utility of $-\langle \boldsymbol{A}, \boldsymbol{C} \rangle$, and $\gamma \boldsymbol{\varepsilon}$ acts as a random utility shock.

**Example: Gumbel Perturbation**. We consider the i.i.d. Gumbel distribution for the perturbation noise and this choice recovers the soft-DTW. Our result is based on the following lemma.

---

[1]For notational simplicity, we omit the explicit dependence on $C$ and write $\boldsymbol{p}$ and $\boldsymbol{s}$.

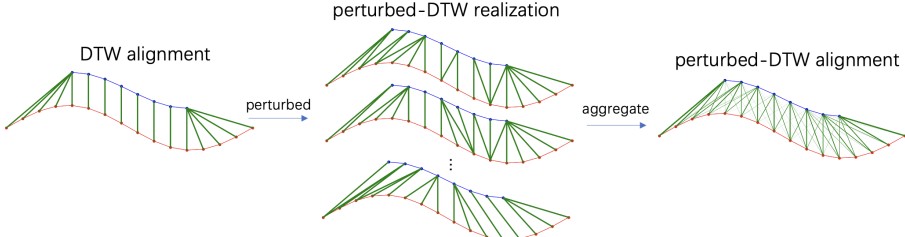

Figure 2: An illustration of perturbed-DTW. The left one is the alignment under the DTW, the middle one is the alignments under different random noise realizations, and the right one can be treated as the aggregation of middle ones and it is also the alignment under soft-DTW.

**Lemma 1.** *If $\varepsilon$ are i.i.d. Gumbel$(-c, 1)$ distributed, where $c \approx 0.5772$ is the Euler-Mascheroni constant, then*

$$\mathbb{E}\Big[ \min\{x_1 - \gamma\varepsilon_1, \dots, x_n - \gamma\varepsilon_n\}\Big] = -\gamma \log \sum_{i=1}^{n} \exp(-\frac{x_i}{\gamma}), \tag{9}$$

This lemma leverages the property of Gumbel distribution, since the minimum of Gumbel variables corresponds to the negative maximum of their negations. Based on this lemma, we can obtain the following result which induces soft-DTW.

**Proposition 1.** *The perturbed-DTW under i.i.d. Gumbel$(-c, 1)$ perturbation is*

$$\text{perturbed-DTW}_\gamma(\boldsymbol{C}) := \mathbb{E}\Big[\min\big\{\langle\mathbf{A}, \mathbf{C}\rangle - \gamma\varepsilon\big\}\Big] = -\gamma \log \sum_{\boldsymbol{A}\in\mathcal{A}_{m,n}} \exp\left(-\frac{\langle\boldsymbol{A},\boldsymbol{C}\rangle}{\gamma}\right).$$

*In addition, the optimal alignment matrix in perturbed-DTW is*

$$P(\boldsymbol{A}; \boldsymbol{C}) = \mathbb{E}\left[\arg\min_{\boldsymbol{A}}\big\{\langle\mathbf{A}, \mathbf{C}\rangle - \gamma\varepsilon\big\}\right] = \frac{\exp\big(-\langle\boldsymbol{A},\boldsymbol{C}\rangle/\gamma\big)}{\sum_{\boldsymbol{A}'\in\mathcal{A}_{m,n}} \exp\big(-\langle\boldsymbol{A}',\boldsymbol{C}\rangle/\gamma\big)}.$$

**Example : Generalized Extreme Value Perturbation**. Beyond the Gumbel case, other perturbation families yield new differentiable relaxations. In particular, we extend the perturbation framework by modeling the random noise with the generalized extreme value (GEV) distribution. Before proceeding, we briefly recall the definition of the GEV distribution.

To formalize this setting, we partition $\mathcal{A}_{m,n}$ into $J$ groups, with the $j$-th group containing $K_j$ elements, such that $\sum_{j=1}^{J} K_j = |\mathcal{A}_{m,n}|$. We index groups by $j$ and denote by $k$ the alignment matrix associated with the $k$-th element of group $j$. Then, the cumulative distribution function (CDF) of GEV joint distribution is

$$F(\varepsilon_{11}, \dots, \varepsilon_{JK_J}) = \exp\left(-\sum_{j=1}^{J}\left[\sum_{k=1}^{K_j}\exp\left(-\frac{\varepsilon_{jk}}{\tau_j}\right)\right]^{\tau_j}\right). \tag{10}$$

where $0 < \tau_\ell \leq 1$ is the similarity parameter. Intuitively, the GEV distribution can be regarded as a correlated multivariate generalization of the Gumbel distribution. By adopting the GEV perturbation, we obtain the following result and denote the new variant as *nested-soft-DTW*, dubbed ns-DTW$_\gamma(\boldsymbol{C})$.

**Theorem 1.** *If the GEV errors are centered to have mean zero (i.e. $\tilde{\varepsilon}_{jk} = \varepsilon_{jk} - c$), the perturbed-DTW under $\tilde{\varepsilon}$ has the following expression:*

$$\text{ns-DTW}_\gamma(\boldsymbol{C}) := \mathbb{E}\Big[\min\big\{\langle\mathbf{A}, \mathbf{C}\rangle - \gamma\tilde{\varepsilon}\big\}\Big] = -\gamma \log\left(\sum_{\ell=1}^{J}\left(\sum_{A\in\ell}\exp\left(-\frac{\langle\boldsymbol{A},\boldsymbol{C}\rangle}{\gamma\tau_\ell}\right)\right)^{\tau_\ell}\right) \tag{11}$$

*Moreover, if $A$ is the kth one in group $j$, the corresponding probability is*

$$P(\boldsymbol{A}; \boldsymbol{C}) = \frac{\left(\sum_{\boldsymbol{A}'\in j}\exp\left(-\frac{\langle\boldsymbol{A}',\boldsymbol{C}\rangle}{\gamma\tau_j}\right)\right)^{\tau_j - 1}}{\sum_{\ell=1}^{J}\left(\sum_{\boldsymbol{A}'\in\ell}\exp\left(-\frac{\langle\boldsymbol{A}',\boldsymbol{C}\rangle}{\gamma\tau_\ell}\right)\right)^{\tau_\ell}} \cdot \exp\left(-\frac{\langle\boldsymbol{A},\boldsymbol{C}\rangle}{\gamma\tau_j}\right)$$

The idea of GEV can also be found in the nest logit model from random utility theory (Train, 2009). Note that perturbed-DTW under GEV perturbation presents groupwise correlation between different $\boldsymbol{A}$ when $0 < \tau_\ell < 1$ and it reduces to soft-DTW when $\tau_\ell = 1, \forall \ell$. Additionally, we can also write it into a variational form.

**Proposition 2.** *The perturbed-DTW under GEV perturbation in Equation (11) can be written into variational form:*

$$\text{ns-DTW}_\gamma(\boldsymbol{C}) = \min_{\boldsymbol{p} \in \Delta^{|\mathcal{A}(m,n)|}} \langle \boldsymbol{p}, \boldsymbol{s} \rangle + \gamma H(\boldsymbol{p}), \tag{12}$$

*where*

$$H(\boldsymbol{p}) = \sum_{\ell=1}^{J} \sum_{m=1}^{K_\ell} \tau_\ell p_{\ell m} \log p_{\ell m} - \sum_{\ell=1}^{J} (\tau_\ell - 1) \left( \sum_{m=1}^{K_\ell} p_{\ell m} \right) \log \left( \sum_{m=1}^{K_\ell} p_{\ell m} \right). \tag{13}$$

We call the regularization term (13) as *nested Shannon entropy* (Fosgerau et al., 2020) and it reduce to Shannon entropy when $\tau_\ell = 1, \forall \ell$. The following proposition characterizes some properties of perturbed-DTW.

**Proposition 3** (Properties of perturbed-DTW)**.** *The following properties hold for perturbed-DTW:*

1. *(Scaling)* $\text{perturbed-DTW}_\gamma(\boldsymbol{C}) = \gamma \, \text{perturbed-DTW}_1(\boldsymbol{C}/\gamma)$.

2. *(Optimal Alignment matrix distribution) The distribution of the alignment matrix is given by* $P(\boldsymbol{A}; \boldsymbol{C}) = \mathbb{E}\left[ \arg\min_{\boldsymbol{A}} \left\{ \langle \boldsymbol{A}, \boldsymbol{C} \rangle - \gamma \boldsymbol{\varepsilon} \right\} \right]$.

3. *(Gradient) The gradient of perturbed-DTW with respect to $C$ is expected alignment matrix* $\boldsymbol{E} = \sum_{\boldsymbol{A} \in \mathcal{A}_{m,n}} \boldsymbol{A} \cdot P(\boldsymbol{A}; \boldsymbol{C})$.

4. *(Asymptotic)* $\text{perturbed-DTW}_\gamma(\boldsymbol{C}) \to \text{DTW}(\boldsymbol{C})$ *and* $\boldsymbol{E} \to \boldsymbol{A}^*$ *as* $\gamma \to 0$.

## 3.2 PERTURBED-DTW COMPUTATION

Analogously, perturbed-DTW faces the computational challenge, as evaluating (8) still requires enumerating over the exponentially large space $\mathcal{A}_{m,n}$. To address this, we adapt the dynamic programming formulation of DTW by replacing the minimum operator in (3) with the perturbed minimum operator. Specifically, we define the *perturbed accumulated cost matrix $\boldsymbol{V}$* recursively as

$$V_{i,j} = \mathbb{E}\left[ \min\left\{ V_{i-1,j-1} - \gamma \varepsilon_{i-1,j-1}, \, V_{i-1,j} - \gamma \varepsilon_{i-1,j}, \, V_{i,j-1} - \gamma \varepsilon_{i,j-1} \right\} \right] + C_{i,j}.$$

This recursive form will end at $V_{m,n}$. In this way, we just need to compute the expectation over three variables in one recursion and set $\text{perturbed-DTW}_\gamma(\boldsymbol{C}) = V_{m,n}$, which makes the computation tractable. We now examine two specific perturbation distributions: the Gumbel and the GEV.

**Example: Gumbel Perturbation**. The dynamic programming formula of perturbed-DTW for Gumbel noise is

$$V_{i,j} = \mathbb{E}\left[ \min\{ V_{i,j-1} - \gamma \varepsilon_{i,j-1}, V_{i-1,j} - \gamma \varepsilon_{i-1,j}, V_{i-1,j-1} - \gamma \varepsilon_{i-1,j-1} \} \right] + C_{i,j}$$
$$= -\gamma \log \left( \exp\left( -\frac{V_{i,j-1}}{\gamma} \right) + \exp\left( -\frac{V_{i-1,j}}{\gamma} \right) + \exp\left( -\frac{V_{i-1,j-1}}{\gamma} \right) \right) + C_{i,j}. \tag{14}$$

which is consistent to soft-DTW. This connection casts perturbed-DTW to differentiable dynamic programming (Mensch & Blondel, 2018) and reveals that soft-DTW is a special case of Gumbel perturbations.

**Proposition 4.** *Define the (random) perturbed cost matrix $\widetilde{\boldsymbol{C}}_\gamma$, where $[\widetilde{\boldsymbol{C}}_\gamma]_{i,j} = C_{i,j} - \gamma \varepsilon_{i,j}$ and $\varepsilon_{i,j}$ is Gumbel$(-c, 1)$ distributed. Then*

$$\text{soft-DTW}_\gamma(\boldsymbol{C}) = \mathbb{E}\left[ \text{DTW}(\widetilde{\boldsymbol{C}}_\gamma) \right]. \tag{15}$$

This shows that soft-DTW equals the expectation of DTW discrepancy for perturbed local cost matrix $\widetilde{C}$ under Gumbel perturbation.

**Example: Generalized Extreme Value Perturbation**. Different from the linear programming formulation of perturbed-DTW in Section 3.1 , here we no longer enumerate alignments in $\mathcal{A}_{m,n}$. Instead, we just need to consider three admissible transition directions $\{\rightarrow, \downarrow, \searrow\}$ at each stage $(i, j)$. When the perturbation vector $\boldsymbol{\varepsilon} = (\varepsilon_{i-1,j-1}, \varepsilon_{i-1,j}, \varepsilon_{i,j-1})$ follows a GEV distribution, three distinct schemes of grouping need to be considered: [2]

$$g_1 = \big\{ \{\rightarrow, \downarrow\}, \{\searrow\} \big\}, \qquad g_2 = \big\{ \{\rightarrow, \searrow\}, \{\downarrow\} \big\}, \qquad g_3 = \big\{ \{\downarrow, \searrow\}, \{\rightarrow\} \big\}.$$

To ease the illustration, we just present $g_1$ and a comprehensive discussion of grouping is deferred to Appendix A.4 . Formally, we divide the directions $\{\rightarrow, \downarrow, \searrow\}$ of warping paths into two groups: $J_1 = \{\downarrow, \rightarrow\}$ and $J_2 = \{\searrow\}$. The perturbation $(\varepsilon_{i-1,j-1}, \varepsilon_{i,j-1}, \varepsilon_{i-1,j})$ follows GEV distribution, then the dynamic programming formula is

$$V_{i,j} = \mathbb{E}\big[\min\{V_{i,j-1} - \gamma\varepsilon_{i,j-1}, V_{i-1,j} - \gamma\varepsilon_{i-1,j}, V_{i-1,j-1} - \gamma\varepsilon_{i-1,j-1}\}\big] + C_{i,j}$$

$$= -\gamma \log\left(\left(\exp(-\tfrac{V_{i,j-1}}{\gamma\tau}) + \exp(-\tfrac{V_{i-1,j}}{\gamma\tau})\right)^{\tau} + \exp\left(-\tfrac{V_{i-1,j-1}}{\gamma}\right)\right) + C_{i,j}. \qquad (16)$$

It is important to clarify that the value $V_{m,n}$ obtained via the dynamic programming recursion (16) is a tractable algorithmic realization, rather than an exact evaluation, of the theoretical ns-DTW defined in (11). This distinction can be understood from two perspectives. First, the global definition in (11) implies a single GEV perturbation of dimension $|\mathcal{A}_{m,n}|$ over the entire alignment space, whereas the DP formulation applies independent, low-dimensional GEV perturbations locally to the three transition directions at each step. Second, regarding the recursive structure: unlike the standard Log-Sum-Exp operator, which satisfies the stability property (i.e., the sum of Gumbel variables follows a Gumbel distribution), the nested application of the generalized operators in (16) does not strictly preserve the form of the global GEV distribution. Despite this theoretical distinction, we refer to the efficient DP output $V_{m,n}$ as ns-DTW throughout this work.

**Effect of Grouping**. The ns-DTW offers greater flexibility in modeling diverse alignment structures by allowing different groups of directions. This enables the algorithm to produce alignments that are skewed towards either the vertical ($\downarrow$) or horizontal ($\rightarrow$) directions, as illustrated in Figure 3.

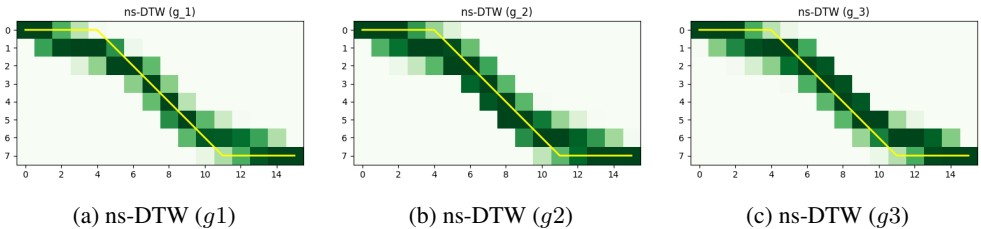

(a) ns-DTW ($g1$)          (b) ns-DTW ($g2$)          (c) ns-DTW ($g3$)

Figure 3: Comparison of warping paths with under different groupings of ns-DTW. The yellow path is depicted as the optimal warping path under DTW.

**Effect of $\tau$**. The flexibility of ns-DTW is also governed by the parameter $\tau$. Setting $\tau = 1$ recovers the standard soft-DTW. Conversely, as $\tau \to 0$, the model becomes increasingly selective: transitions with higher accumulated costs receive vanishing weights, while those with lower costs are emphasized. This mechanism introduces a structural skew toward the optimal path. As illustrated in Figure 4, decreasing $\tau$ progressively enlarges the expected warping path toward the direction of lower cost (e.g., the vertical direction $\downarrow$), allowing the model to adaptively prune high-cost deviations.

Proposition 3 reveals that the expected alignment pathes of perturbed-DTW, which is exactly the expected alignment matrix $\boldsymbol{E}$ under the distribution $P(\boldsymbol{A}; \boldsymbol{C})$. However, this general alignment matrix distribution is hard to compute for general perturbation $\varepsilon$. Specifically, let transition probability tensor $\mathbf{G} \in (0,1]^{m \times n \times 3}$. For any given $(i, j)$, $\mathbf{G}_{i,j,:} \in \mathbb{R}^3$ specifies a stochastic alignment policy over feasible actions $\{\rightarrow, \downarrow, \searrow\}$, so that $P(A_{i,j-1} = 1) = \mathbf{G}_{i,j,1}$, $P(A_{i-1,j} = 1) = \mathbf{G}_{i,j,2}$, and $P(A_{i-1,j-1} = 1) = \mathbf{G}_{i,j,3}$. The transition probability tensor $\mathbf{G}$ can be computed as

$$\mathbf{G}_{i,j,:} = \mathbb{E}\big[\arg\min\{V_{i,j-1} - \gamma\varepsilon_{i,j-1}, V_{i-1,j} - \gamma\varepsilon_{i-1,j}, V_{i-1,j-1} - \gamma\varepsilon_{i-1,j-1}\}\big]$$

$$= -\gamma\nabla\log\left(\left(\exp(-\tfrac{V_{i,j-1}}{\gamma\tau}) + \exp(-\tfrac{V_{i-1,j}}{\gamma\tau})\right)^{\tau} + \exp\left(-\tfrac{V_{i-1,j-1}}{\gamma}\right)\right). \qquad (17)$$

---

[2]It can be revealed that grouping all three directions together—or placing each direction in its own group— yields the classical soft-DTW recursion.

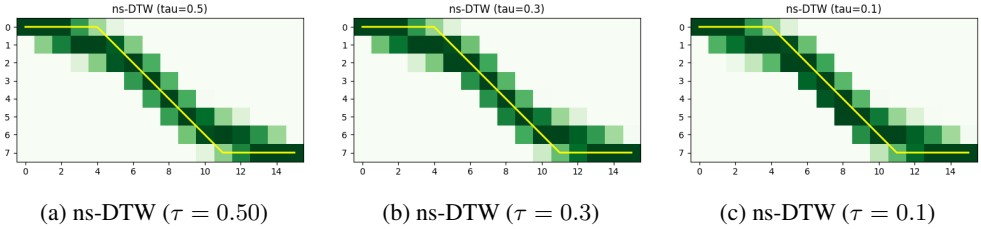

(a) ns-DTW ($\tau = 0.50$)  (b) ns-DTW ($\tau = 0.3$)  (c) ns-DTW ($\tau = 0.1$)

Figure 4: Comparison of warping paths with different temperature parameters $\tau$ of ns-DTW.

where the gradient is taken with respect to $\{V_{i,j-1}, V_{i-1,j}, V_{i-1,j-1}\}$. Then the expected alignment matrix is computed as

$$E_{i,j} = \boldsymbol{G}_{i,j+1,1} E_{i,j+1} + \boldsymbol{G}_{i+1,j,2} E_{i+1,j} + \boldsymbol{G}_{i+1,j+1,3} E_{i+1,j+1}.$$

This recursive formula can be treated as the differentiable dynamic programming with nested Shannon entropy. The ns-DTW allows the model to control the intensity (skewness) toward directions associated with lower accumulated costs, by using $\tau$ and grouping schemes to introduce skewness. By promoting a broader mixture over feasible paths, ns-DTW can approximate the true alignment cost more effectively than soft-DTW. Algorithm 1 presents the pseudocode-for computing ns-DTW and its transition probability tensor. A more detailed discussion is provided in the appendix.

---

**Algorithm 1** ns-DTW and transition probability tensor computation

---

**Require:** Cost matrix $\boldsymbol{C} \in \mathbb{R}^{m \times n}$, $\gamma \geq 0$, $0 < \tau \leq 1$
 1: Initialize: $V_{i,0} \leftarrow \infty$ for all $i$; $V_{0,j} \leftarrow \infty$ for all $j$; $V_{0,0} \leftarrow 0$
 2: **for** $i \in [1, \ldots, m]$ and $j \in [1, \ldots, n]$ **do**
 3:   Compute $V_{i,j}$ via (16)
 4:   Compute $\boldsymbol{G}_{i,j,:}$ via (17)
 5: **end for**
 6: **Return**:
 7: ns-DTW$_\gamma(\boldsymbol{C}) = V_{m,n} \in \mathbb{R}$; $\mathbf{G} \in (0,1]^{m \times n \times 3}$

---

## 4 APPLICATIONS

In this section, we conduct experiments using the UCR Time Series Classification Archive (Chen et al., 2015) . We consider a subset of the archive containing 47 datasets for average, classification[3] and clustering tasks. We report a summary of our results in the manuscript, with full details provided in the appendix.

### 4.1 AVERAGING

We investigate the problem of computing Fréchet mean of time series with respect to ns-DTW. Given a collection of time series: $\mathbf{y}_1, \ldots, \mathbf{y}_M$, our goal is to compute a barycenter $\mathbf{x}$ that minimizes the total ns-DTW discrepancy:

$$\min_{\mathbf{x} \in \mathbb{R}^{p \times m}} \sum_{i=1}^{M} \text{ns-DTW}(\boldsymbol{C}(\mathbf{x}, \mathbf{y}_i))$$

We evaluate the quality of the barycenters in terms of DTW discrepancy and compare ns-DTW with several established baselines, including subgradient method (Schultz & Jain, 2018), DBA (Petitjean et al., 2011) and soft-DTW (Cuturi & Blondel, 2017) methods.

We evaluate the methods based on three direction grouping schemes: $\{g_1, g_2, g_3\}$, parameters $\tau \in \{0.80, 0.85, 0.90, 0.95\}$ and $\gamma \in \{0.1, 0.01, 0.001, 0.0001\}$. Table 1 summarizes the barycenter

---

[3]As some datasets contain only one class, we use 43 datasets for the classification task.

averaging results across varying $\gamma$. By selecting the optimal grouping scheme $g_i$ and parameter $\tau$, ns-DTW demonstrates robust performance improvements. Specifically, it achieves a lower objective value than the Subgradient method on **100%** of datasets and DBA on **97.87%**. Furthermore, when compared against the strongest baseline, soft-DTW (both methods tuned for optimal $\gamma$), ns-DTW yields superior results on **74.47%** of the datasets. Qualitative results on the *Beef* dataset are visualized in Figure 5. Comprehensive dataset-wise results are provided in Appendix B.1.

Table 1: Percentage of the datasets on which the proposed ns-DTW barycenter is achieving lower DTW loss than competing methods.

|                    | subgradient | DBA     | soft-DTW |
| ------------------ | ----------- | ------- | -------- |
| $\gamma = 0.1$     | 68.09%      | 46.81%  | 36.17%   |
| $\gamma = 0.01$    | 80.85%      | 72.34%  | 59.57%   |
| $\gamma = 0.001$   | 95.74%      | 87.23%  | 80.85%   |
| $\gamma = 0.0001$  | 100.00%     | 91.49%  | 91.49%   |

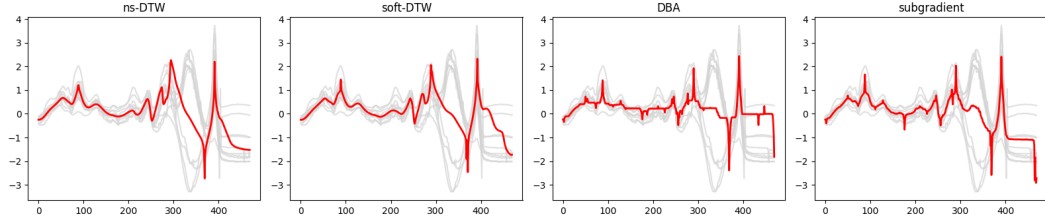

Figure 5: Comparison of barycenters obtained by ns-DTW, soft-DTW and DBA methods on the UCR time series *Beef* dataset, using Euclidean averaging for initialization.

## 4.2 CLASSIFICATION

**Nearest Centroid Classifier**. We evaluated time series classification performance using the *Nearest Centroid Classifier* (NCC), where class centroids were computed as barycenters using the respective averaging algorithms. We employed a 50%/25%/25% train-validation-test split, with the smoothing parameter $\gamma \in \{0.1, 0.01, 0.001, 0.0001\}$ selected via cross-validation.

Overall, ns-DTW demonstrated superior performance, achieving equal or higher accuracy compared to Subgradient methods on **93.02%** of datasets, DBA on **88.37%**, and soft-DTW on **86.05%**. Figure 6 presents the pairwise performance comparison between ns-DTW and baselines for NCC classification.

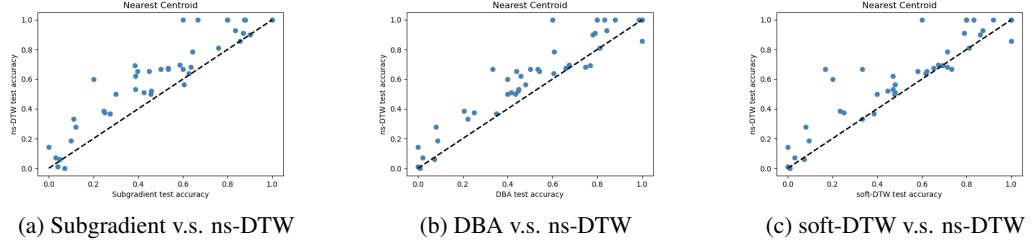

(a) Subgradient v.s. ns-DTW      (b) DBA v.s. ns-DTW      (c) soft-DTW v.s. ns-DTW

Figure 6: Points above the diagonal indicate datasets where the ns-DTW outperforms (a) subgradient; (b)DBA; (c) soft-DTW in nearest centroid classifier.

**1NN Classifier**. We also evaluated the *1-Nearest Neighbor (1NN)* classifier, where test samples are assigned to the class of the nearest training example based on the minimized DTW discrepancy. The data splits and cross-validation for $\gamma$ remained consistent with the NCC experiments.

Overall, ns-DTW achieved equal or higher accuracy compared to DBA on **88.37%** of datasets and soft-DTW on **86.05%**.

## 4.3 CLUSTERING

We study the $k$-means clustering task: the distances between each time series are ns-DTW discrepancy and the centroids of each class are their barycenters. Formally, the objective is to find clusters $C_1, \ldots, C_K$ that minimize total within-cluster ns-DTW discrepancy:

$$\min_{C_1,\ldots,C_K} \sum_{k=1}^{K} \sum_{i,j \in C_k} \text{ns-DTW}(\boldsymbol{C}(\mathbf{y}_i, \mathbf{y}_j)).$$

Overall, our ns-DTW outperforms DBA on **100.00%** and soft-DTW on **76.60%**. Figure 7 presents the clustering results on the *CBF* dataset and the complete results are shown in Appendix B.4.

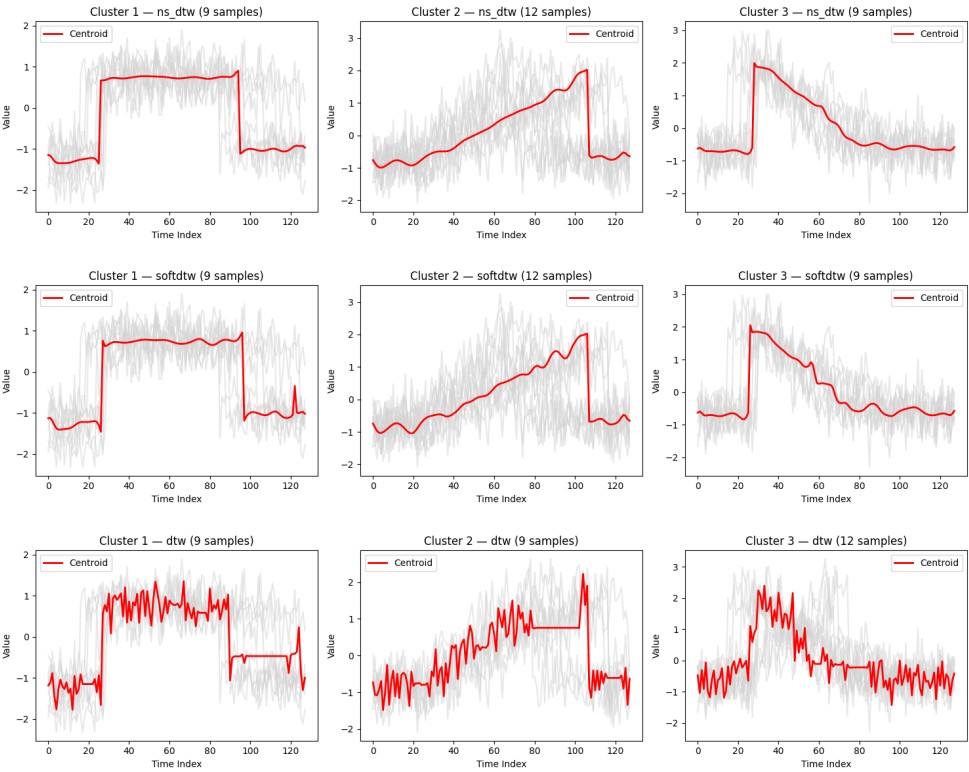

Figure 7: Comparison of clustering results obtained by ns-DTW (first row), soft-DTW (second row) and DBA (third row) methods on the UCR time series *CBF* dataset, initialized using Euclidean averaging.

## 5 CONCLUSION

We introduced *perturbed-DTW*, a probabilistic framework that makes DTW differentiable by adding random perturbations to the warping costs. This perspective recovers soft-DTW under Gumbel perturbation, providing a natural probabilistic interpretation. Extending to the generalized extreme value family leads to *nested soft-DTW*, which enables tunable skewness in alignments and greater modeling flexibility.

Experiments on barycenter computation and clustering demonstrate competitive performance over existing methods. Looking forward, extending perturbed-DTW to divergence-based formulations and to broader classes of perturbed dynamic programming operators offers promising directions for future work.

ACKNOWLEDGMENTS

Xiangqian Sun was supported by grant from XJTLU Research Development Fund [RDF-23-02-052]. Chaoqun Wang was supported by grant from XJTLU Research Development Fund [RDF-24-02-033]. Wei Zhang was partly supported by grants from the National Natural Science Foundation of China [Grants 72401257, 72431010 , 71821002];

ETHICS STATEMENT

This research does not raise any potential violations of the ICLR Code of Ethics. In particular, it does not involve human subjects, dataset release practices, or potentially harmful insights, methodologies, or applications. It also raises no concerns regarding conflicts of interest or sponsorship, discrimination, bias, or fairness, nor does it present issues related to privacy, security, legal compliance, or research integrity (e.g., IRB approval, documentation, or research ethics).

REPRODUCIBILITY STATEMENT

The datasets used in this research are publicly available at `https://www.cs.ucr.edu/~eamonn/time_series_data/`, and the code for reproducing our results is provided in the supplementary material.

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

# A  PROOFS AND COMPUTATIONS

## A.1  PROOF OF THEOREM 1

We first show the following result.

**Lemma 2.** *Let*

$$M := \max_{j \in [J], \, k \in [K_j]} \{\mu_{jk} + \varepsilon_{jk}\},$$

*with $\varepsilon$ following the GEV joint distribution*

$$F(\varepsilon_{11}, \ldots, \varepsilon_{JK_J}) = \exp\left(-\sum_{j=1}^{J} \Big[\sum_{k=1}^{K_j} \exp\Big(-\frac{\varepsilon_{jk}}{\tau_j}\Big)\Big]^{\tau_j}\right).$$

*Then*

$$\mathbb{E}[M] = \left(\sum_{j=1}^{J} \left(\sum_{k=1}^{K_j} \exp\left(\frac{\mu_{jk}}{\tau_j}\right)\right)^{\tau_j}\right).$$

*Proof.* For any $t \in \mathbb{R}$,

$$
\begin{aligned}
\Pr(M \le t) &= \Pr(\varepsilon_{jk} \le t - \mu_{jk} \; \forall j, k) \\
&= F(t - \mu_{11}, \ldots, t - \mu_{JK_J}) \\
&= \exp\left(-\sum_{j=1}^{J} \Big[\sum_{k=1}^{K_j} \exp\Big(-\frac{t - \mu_{jk}}{\tau_j}\Big)\Big]^{\tau_j}\right) \\
&= \exp\left(-\sum_{j=1}^{J} \Big[e^{-t/\tau_j} \sum_{k=1}^{K_j} e^{\mu_{jk}/\tau_j}\Big]^{\tau_j}\right) \\
&= \exp\left(-e^{-t} \sum_{j=1}^{J} \Big(\sum_{k=1}^{K_j} e^{\mu_{jk}/\tau_j}\Big)^{\tau_j}\right).
\end{aligned}
$$

Define

$$\Omega(\mu) = \log\left(\sum_{j=1}^{J} \Big(\sum_{k=1}^{K_j} e^{\mu_{jk}/\tau_j}\Big)^{\tau_j}\right).$$

Then

$$\Pr(M \le t) = \exp\big(-\exp\big(-(t - \Omega(\mu))\big)\big),$$

so $M$ is Gumbel distributed with location $\Omega(\mu)$ and scale 1. By property of Gumbel distribution, we know that

$$\mathbb{E}[M] = \Omega(\mu) + c,$$

where $c \approx 0.5772$ is the Euler-Mascheroni constant. If the GEV errors are centered to have mean zero (i.e. $\tilde{\varepsilon}_{jk} = \varepsilon_{jk} - c$), then

$$\mathbb{E}\left[\max_{j,k}\{\mu_{jk} + \tilde{\varepsilon}_{jk}\}\right] = \Omega(\mu).$$

$\square$

Next, we give the proof of Theorem 1.

*Proof.* If the GEV errors are centered to have mean zero (i.e. $\tilde{\varepsilon}_{jk} = \varepsilon_{jk} - c$),

$$
\begin{aligned}
\text{perturbed-DTW}_\gamma(C) &:= \mathbb{E}\left[ \min_{A \in \mathcal{A}_{m,n}} \left\{ \langle A, C \rangle - \gamma \tilde{\varepsilon} \right\} \right] \\
&= -\gamma \mathbb{E}\left[ \max_{A \in \mathcal{A}_{m,n}} \left\{ -\frac{\langle A, C \rangle}{\gamma} + \tilde{\varepsilon} \right\} \right] \\
&= -\gamma \log \left( \sum_{\ell=1}^{J} \left( \sum_{A \in J} \exp\left( -\frac{\langle A, C \rangle}{\gamma \tau_\ell} \right) \right)^{\tau_\ell} \right),
\end{aligned}
$$

where the last equality comes from Lemma 2. Denote $s_{jk} = \langle A, C \rangle$ if $A$ is the $k$th one in group $j$. The probability distritution of alignment matrix is

$$
\begin{aligned}
P_{jk} &= \frac{\partial}{\partial s_{jk}} \left[ -\gamma \log \left( \sum_{\ell=1}^{J} \left( \sum_{A \in J} \exp\left( -\frac{s_{jk}}{\gamma \tau_\ell} \right) \right)^{\tau_\ell} \right) \right] \\
&= \frac{\left( \sum_{m=1}^{K_j} \exp\left( \frac{s_{jm}}{\gamma \tau_j} \right) \right)^{\tau_j - 1}}{\sum_{\ell=1}^{J} \left( \sum_{m=1}^{K_\ell} \exp\left( \frac{s_{\ell m}}{\gamma \tau_\ell} \right) \right)^{\tau_\ell}} \exp\left( \frac{s_{jk}}{\gamma \tau_j} \right),
\end{aligned}
$$

as desired. $\qquad\square$

## A.2 PROOF OF PROPOSITION 2

*Proof.* For simplicity, we just show the case $\gamma = 1$. The Lagrangian function is defined as

$$
L(\boldsymbol{p}, \lambda) = \langle \boldsymbol{p}, \boldsymbol{s}(C) \rangle + H(\boldsymbol{p}) + \lambda \left( \sum_\ell \sum_m p_{\ell m} - 1 \right). \tag{18}
$$

The optimality condition gives

$$
\begin{cases} s_{\ell m} + \tau_\ell (1 + \log p_{\ell m}) - (\tau_\ell - 1)(1 + \sum_m p_{\ell m}) + \lambda = 0, & \forall \ell, m \\ \sum_\ell \sum_m p_{\ell m} = 1. \end{cases} \tag{19}
$$

Equation 19 implies

$$
p_{\ell m} = \exp\left( \frac{-s_{\ell m} - 1 - \lambda}{\tau_\ell} + \frac{\tau_\ell - 1}{\tau_\ell} \log \sum_m p_{\ell m} \right). \tag{20}
$$

With a little abuse of notations, we denote $p_\ell = \sum_m p_{\ell m}$. Then

$$
p_\ell = \sum_m p_{\ell m} = \sum_m \exp\left( \frac{-s_{\ell m} - 1 - \lambda}{\tau_\ell} + \frac{\tau_\ell - 1}{\tau_\ell} \log \sum_m p_{\ell m} \right) \tag{21}
$$

$$
= \sum_m \exp\left( \frac{-s_{\ell m} - 1 - \lambda}{\tau_\ell} + \frac{\tau_\ell - 1}{\tau_\ell} \log p_\ell \right) \tag{22}
$$

$$
\implies \frac{1}{\tau_\ell} \log p_\ell = \log \sum_m \exp\left( \frac{-s_{\ell m} - 1 - \lambda}{\tau_\ell} \right). \tag{23}
$$

By leveraging the equality condition, we have

$$
1 = \sum_\ell p_\ell = \sum_\ell \left( \sum_m \exp\left( \frac{-s_{\ell m} - 1 - \lambda}{\tau_\ell} \right) \right)^{\tau_\ell} = \sum_\ell \left( \sum_m \exp\left( \frac{-s_{\ell m}}{\tau_\ell} \right) \right)^{\tau_\ell} / \exp(1 + \lambda) \tag{24}
$$

$$
\implies \lambda = \log \left( \sum_\ell \left( \sum_m \exp\left( \frac{-s_{\ell m}}{\tau_\ell} \right) \right)^{\tau_\ell} \right) - 1. \tag{25}
$$

By substituting (23) and (25) into (20), we have

$$
\begin{aligned}
p_{\ell m} &= \exp\left(\frac{-s_{\ell m}}{\tau_\ell} - \frac{1}{\tau_\ell} - \frac{\lambda}{\tau_\ell} + (\tau_\ell - 1)\log\sum_m \exp\left(\frac{-s_{\ell m}}{\tau_\ell}\right) - 1 + \frac{1}{\tau_\ell} - \lambda + \frac{\lambda}{\tau_\ell}\right) \\
&= \exp\left(\frac{-s_{\ell m}}{\tau_\ell} + (\tau_\ell - 1)\log\sum_m \exp\left(\frac{-s_{\ell m}}{\tau_\ell}\right) - \log\left(\sum_\ell\left(\sum_m \exp(\frac{-s_{\ell m}}{\tau_\ell})\right)^{\tau_\ell}\right)\right).
\end{aligned}
$$
(26)

The proof completes by plugging (26) into (6). $\qquad\square$

### A.3  PROOF OF PROPOSITION 3

The fourth property is straightforward; here, we present only the first three.

*Proof.*    1. By definition of perturbed-DTW, we have

$$
\begin{aligned}
\text{perturbed-DTW}_\gamma(\boldsymbol{C}) &= \mathbb{E}_{\boldsymbol{\varepsilon}\sim\mathbb{P}}\left[\min\left\{\langle \mathbf{A}, \mathbf{C}\rangle - \gamma\varepsilon\right\}\right] \\
&= \gamma\mathbb{E}_{\boldsymbol{\varepsilon}\sim\mathbb{P}}\left[\min\left\{\frac{1}{\gamma}\langle \mathbf{A}, \mathbf{C}\rangle - \varepsilon\right\}\right] \\
&= \gamma\,\text{perturbed-DTW}_1(\boldsymbol{C}/\gamma).
\end{aligned}
$$

2. The optimal alignment matrix $\boldsymbol{A}$ follows

$$
\begin{aligned}
P(\boldsymbol{A};\boldsymbol{C}) &= P\left(\langle \boldsymbol{A}, \boldsymbol{C}\rangle - \gamma\varepsilon \le \langle \boldsymbol{A}', \boldsymbol{C}\rangle - \gamma\varepsilon', \forall \boldsymbol{A}', \varepsilon'\right) \\
&= \mathbb{E}\left[\arg\min_{\boldsymbol{A}}\left\{(\langle \mathbf{A}, \mathbf{C}\rangle) - \gamma\varepsilon\right\}\right].
\end{aligned}
$$

3. By the Williams–Daly–Zachary theorem (McFadden, 1981),

$$
\begin{aligned}
\nabla_{\boldsymbol{C}}\,\text{perturbed-DTW}_\gamma(\boldsymbol{C}) &= \left\langle \mathbb{E}\left[\arg\min_{\langle \boldsymbol{A}, \boldsymbol{C}\rangle}\left\{\langle \mathbf{A}, \mathbf{C}\rangle - \gamma\varepsilon\right\}\right], \nabla_{\boldsymbol{C}}\langle \boldsymbol{A}, \boldsymbol{C}\rangle \right\rangle \\
&= \sum_{\boldsymbol{A}\in\mathcal{A}_{m,n}} \mathbb{E}\left[\arg\min_{\boldsymbol{A}\in\mathcal{A}_{m,n}}\left\{\langle \mathbf{A}, \mathbf{C}\rangle - \gamma\varepsilon\right\}\right]\cdot\boldsymbol{A} \\
&= \sum_{\boldsymbol{A}\in\mathcal{A}_{m,n}} P(\boldsymbol{A};\boldsymbol{C})\cdot\boldsymbol{A} = \boldsymbol{E}.
\end{aligned}
$$

$\qquad\square$

### A.4  GENERAL FORMULATION OF NS-DTW

As discussed earlier, ns-DTW can be viewed as a generalized perturbed variant of soft-DTW obtained by replacing the Gumbel perturbation with a GEV perturbation. Note that the GEV distribution can be regarded as multivariate generalization of the Gumbel distribution with groupwise correlation. This substitution introduces not only a hyper-parameter $\tau$ but also multiple schemes of direction groupings. The dynamic programming formula for general perturbed-DTW is

$$
V_{i,j} = \mathbb{E}\left[\min\left\{V_{i-1,j-1} - \gamma\varepsilon_{i-1,j-1},\; V_{i-1,j} - \gamma\varepsilon_{i-1,j},\; V_{i,j-1} - \gamma\varepsilon_{i,j-1}\right\}\right] + C_{i,j}.
$$

When the perturbation vector $\boldsymbol{\varepsilon} = (\varepsilon_{i-1,j-1}, \varepsilon_{i-1,j}, \varepsilon_{i,j-1})$ follows a GEV distribution, three distinct grouping schemes need to be considered: [4]

$$
g_1 = \left\{\{\rightarrow, \downarrow\}, \{\searrow\}\right\}, \qquad g_2 = \left\{\{\rightarrow, \searrow\}, \{\downarrow\}\right\}, \qquad g_3 = \left\{\{\downarrow, \searrow\}, \{\rightarrow\}\right\}.
$$

---

[4]It can be revealed that grouping all three directions together—or placing each direction in its own group—yields the classical soft-DTW recursion.

Therefore, the general formulas for dynamic-programming recursions for different grouping schemes are

$$g_1: \quad V_{i,j} = -\gamma \log\left[\left(e^{-V_{i,j-1}/(\gamma\tau)} + e^{-V_{i-1,j}/(\gamma\tau)}\right)^{\tau} + e^{-V_{i-1,j-1}/\gamma}\right] + C_{i,j}, \qquad (27)$$

$$g_2: \quad V_{i,j} = -\gamma \log\left[\left(e^{-V_{i,j-1}/(\gamma\tau)} + e^{-V_{i-1,j-1}/(\gamma\tau)}\right)^{\tau} + e^{-V_{i-1,j}/\gamma}\right] + C_{i,j}, \qquad (28)$$

$$g_3: \quad V_{i,j} = -\gamma \log\left[\left(e^{-V_{i-1,j}/(\gamma\tau)} + e^{-V_{i-1,j-1}/(\gamma\tau)}\right)^{\tau} + e^{-V_{i,j-1}/\gamma}\right] + C_{i,j}. \qquad (29)$$

## A.5   NS-DTW COMPUTATION

To streamline the presentation, we focus on the ns-DTW computation under the first scheme of grouping $g_1$. The transition probabilities and gradients for the remaining groupings can be derived analogously. The dynamic programming formula of perturbed-DTW under GEV perturbation in $g_1$ is

$$V_{i,j} = \mathbb{E}\left[\min\{V_{i,j-1} + C_{i,j} - \gamma\varepsilon_{i,j-1}, V_{i-1,j} + C_{i,j} - \gamma\varepsilon_{i-1,j}, V_{i-1,j-1} + C_{i,j} - \gamma\varepsilon_{i-1,j-1}\}\right]$$

$$= -\gamma \log\left(\left(\exp(-\tfrac{V_{i,j-1}}{\gamma\tau}) + \exp(-\tfrac{V_{i-1,j}}{\gamma\tau})\right)^{\tau} + \exp\left(-\tfrac{V_{i-1,j-1}}{\gamma}\right)\right) + C_{i,j}. \qquad (30)$$

Therefore, the transition probability is

$$P(A_{i,j-1} = 1) = \mathbf{G}_{i,j,1} = \frac{\left(\exp(-\tfrac{V_{i,j-1}}{\gamma\tau}) + \exp(-\tfrac{V_{i-1,j}}{\gamma\tau})\right)^{\tau-1}}{\left(\exp(-\tfrac{V_{i,j-1}}{\gamma\tau}) + \exp(-\tfrac{V_{i-1,j}}{\gamma\tau})\right)^{\tau} + \exp\left(-\tfrac{V_{i-1,j-1}}{\gamma}\right)} \cdot \exp\left(-\tfrac{V_{i,j-1}}{\gamma\tau}\right),$$

$$P(A_{i-1,j} = 1) = \mathbf{G}_{i,j,2} = \frac{\left(\exp(-\tfrac{V_{i,j-1}}{\gamma\tau}) + \exp(-\tfrac{V_{i-1,j}}{\gamma\tau})\right)^{\tau-1}}{\left(\exp(-\tfrac{V_{i,j-1}}{\gamma\tau}) + \exp(-\tfrac{V_{i-1,j}}{\gamma\tau})\right)^{\tau} + \exp\left(-\tfrac{V_{i-1,j-1}}{\gamma}\right)} \cdot \exp\left(-\tfrac{V_{i-1,j}}{\gamma\tau}\right),$$

$$P(A_{i-1,j-1} = 1) = \mathbf{G}_{i,j,3} = \frac{\exp\left(-\tfrac{V_{i-1,j-1}}{\gamma}\right)}{\left(\exp(-\tfrac{V_{i,j-1}}{\gamma\tau}) + \exp(-\tfrac{V_{i-1,j}}{\gamma\tau})\right)^{\tau} + \exp\left(-\tfrac{V_{i-1,j-1}}{\gamma}\right)}.$$

Then the expected alignment matrix is computed by

$$E_{i,j} = \mathbf{G}_{i,j+1,1}E_{i,j+1} + \mathbf{G}_{i+1,j,2}E_{i+1,j} + \mathbf{G}_{i+1,j+1,3}E_{i+1,j+1}.$$

Algorithm 2 presents the pseudocode for computing gradient of ns-DTW.

---

**Algorithm 2** ns-DTW gradient computation

---

**Require:** $\mathbf{G} \in (0,1]^{m \times n \times 3}$ (Algorithm 1)
1: Initialize: $E_{m+1,:} \leftarrow 0$, $E_{:,n+1} \leftarrow 0$, $E_{m+1,n+1} \leftarrow 1$
2: Initialize: $\mathbf{G}_{m+1,:,:} \leftarrow (0,0,0)$, $\mathbf{G}_{:,n+1,:} \leftarrow (0,0,0)$, $\mathbf{G}_{m+1,n+1,:} \leftarrow (0,1,0)$
3: **for** $i \in [m, \ldots, 1], j \in [n, \ldots, 1]$, **do**
4:     $E_{i,j} \leftarrow \mathbf{G}_{i,j+1,1} \cdot E_{i,j+1} + \mathbf{G}_{i+1,j+1,2} \cdot E_{i+1,j+1} + \mathbf{G}_{i+1,j,3} \cdot E_{i+1,j}$
5: **end for**
6: **Return:** $\nabla_C \text{ns-DTW}_\gamma(C) = \mathbf{E} \in (0,1]^{m \times n}$

---

Consider the warping cost $\mathbf{C} \in \mathbb{R}^{m \times n}$, to compute the value of ns-DTW, Algorithm 1 requires $O(mn)$ operations and $O(mn)$ storage cost as well. This is as same as the soft-DTW. However, if we consider different groupings, in other words, dividing directions into two groups , the computational cost would be three times as soft-DTW. (since soft-DTW can be treated as one special grouping of ns-DTW).

## A.6 Implementation Details and Experimental Setup

**Software Environment.** All experiments were implemented in **Python 3.9**. The core logic relies on `tslearn` (v0.6.4) for time series operations, `numpy` (v2.0.2) and `pandas` (v2.3.3) for data manipulation, and `scikit-learn` (v1.6.1) for evaluation metrics.

**Hyperparameter Configuration.** To ensure reproducibility and fair comparison, we standardized the search space and initialization protocols across all datasets. We utilized Euclidean averaging for initialization and set a maximum budget of 30 iterations for the barycenter computation, observing convergence in most cases within this limit. The specific hyperparameter ranges and grouping schemes for ns-DTW are detailed in Table 2.

Table 2: Summary of experimental settings and hyperparameter search spaces.

| Parameter | Value / Definition |
|---|---|
| **Software** | Python 3.9, tslearn 0.6.4, numpy 2.0.2, sklearn 1.6.1 |
| **Parameter** ($\tau$) | $\{0.80,\ 0.85,\ 0.90,\ 0.95\}$ |
| **Smoothing** ($\gamma$) | $\{0.1, 0.01, 0.001, 0.0001\}$ |
| **Grouping Schemes** | $g_1 = \{\{\rightarrow, \downarrow\}, \{\searrow\}\}$ 
 $g_2 = \{\{\rightarrow, \searrow\}, \{\downarrow\}\}$ 
 $g_3 = \{\{\downarrow, \searrow\}, \{\rightarrow\}\}$ |
| **Initialization** | Euclidean Averaging |
| **Max Iterations** | 30 |

We clarify the hyperparameter selection procedure for two types of tasks: averaging and clustering( unsupervised), and classification (supervisd). Detailed results are presented in Appendix B.

**Averaging and Clustering Tasks:** For these tasks, we use a grid search to evaluate combinations of grouping schemes ($g_i \in \{g_1, g_2, g_3\}$), parameter $\tau \in \{0.80, 0.85, 0.90, 0.95\}$ and the parameter $\gamma \in \{0.1, 0.01, 0.001, 0.0001\}$. In other words, we evaluate all datasets for every possible combination of $(g_i, \tau, \gamma)$.
To examine the **effect of** $g_i$, we fixed each grouping scheme and then found the best $\tau$ and $\gamma$ combination within the grid search that yielded the lowest DTW losses. These results are presented in the last three columns of Tables 3 and 6.
To examine the **effect of** $\tau$, we collected results for fixed $\tau$ and $g_i$, then selected the $\gamma$ that achieved the lowest DTW losses. The detailed results for averaging are in Appendix C (Table 7 for $g_1$, Table 8 for $g_2$, and Table 9 for $g_3$). For clustering, results are in Appendix E (Table 13 for $g_1$, Table 14 for $g_2$, and Table 15 for $g_3$).

**Classification Task:** For classification, we still perform a grid search over $g_i \in \{g_1, g_2, g_3\}$ and $\tau \in \{0.80, 0.85, 0.90, 0.95\}$. However, for each $\{g_i, \tau\}$ pair, $\gamma$ is selected via cross-validation on the training set. This approach allows us to specifically evaluate the effects of different grouping schemes and $\tau$ while ensuring $\gamma$ is optimally tuned for each combination.
To examine the **effect of** $g_i$, we fixed each grouping scheme and then found the best $\tau$ (with $\gamma$ determined by cross-validation ) within the grid search that yielded the highest classification accuracy. The results of $g_i$ on classification performance are presented in the last three columns of Table 4.
To examine the **effect of** $\tau$, we collected results for fixed $\tau$ and $g_i$, then selected the $\gamma$ that achieved the highest classification accuracy. Further results showing the effects of different groupings and $\tau$ with cross-validated $\gamma$ are provided in Appendix D (Table 10 for $g_1$, Table 11 for $g_2$, and Table 12 for $g_3$) .

# B  RESULTS

## B.1  AVERAGING

Table 3: UCR Barycenter DTW Losses

| Dataset | subgradient | DBA | soft-DTW | ns-DTW ($g_1$) | ns-DTW ($g_2$) | ns-DTW ($g_3$) |
|---|---|---|---|---|---|---|
| Adiac | 0.2594 | 0.2576 | 0.2552 | **0.2551** | 0.2553 | 0.2552 |
| ArrowHead | 1.2413 | 1.1893 | **1.1440** | 1.1646 | 1.1640 | 1.1545 |
| Beef | 8.4287 | 4.1666 | **4.0666** | 4.1005 | 4.1025 | 4.1015 |
| BeetleFly | 4.9672 | 4.5245 | 4.2886 | 4.1907 | 4.1909 | **4.1905** |
| BirdChicken | 4.9910 | 2.4679 | **2.3858** | 2.3930 | 2.3921 | 2.3938 |
| CBF | 4.3019 | 3.8861 | **3.4473** | 3.7857 | 3.8222 | 3.7833 |
| Car | 0.6908 | 0.6703 | **0.5948** | 0.5954 | 0.5964 | 0.5963 |
| ChlorineConcentration | 3.6285 | 3.4941 | 3.4297 | **3.4047** | 3.4061 | 3.4091 |
| CinCECGTorso | 17.9134 | 8.8686 | **8.3582** | 8.3653 | 8.3597 | 8.3599 |
| Coffee | 0.6249 | 0.5950 | **0.5872** | 0.5890 | 0.5892 | 0.5893 |
| Computers | 17.2013 | 14.8046 | **14.6065** | 14.6314 | 14.6228 | 14.6203 |
| CricketX | 7.9406 | 5.9852 | 5.7930 | 5.6919 | 5.7399 | **5.6772** |
| CricketY | 6.3425 | 5.7147 | **5.5186** | 5.5646 | 5.5598 | 5.5645 |
| CricketZ | 6.2965 | 5.4289 | 5.3826 | 5.3563 | 5.3752 | **5.3454** |
| DiatomSizeReduction | 0.2529 | 0.2357 | 0.2282 | **0.2278** | 0.2278 | 0.2278 |
| DistalPhalanxOutlineAgeGroup | 0.7220 | 0.7073 | 0.7043 | **0.7036** | 0.7038 | 0.7038 |
| DistalPhalanxOutlineCorrect | 0.6972 | 0.6670 | **0.6658** | 0.6662 | 0.6662 | 0.6662 |
| DistalPhalanxTW | 0.3622 | 0.3478 | **0.3448** | 0.3458 | 0.3458 | 0.3458 |
| ECG200 | 2.7862 | 2.7185 | 2.6750 | **2.6470** | 2.6537 | 2.6704 |
| ECG5000 | 2.2571 | 2.2061 | 2.2012 | **2.1968** | 2.2024 | 2.2023 |
| ECGFiveDays | 2.7898 | 2.5564 | 2.5618 | 2.5275 | **2.4869** | 2.5243 |
| Earthquakes | 12.2798 | 10.7766 | **10.6706** | 10.7192 | 10.7052 | 10.7222 |
| FaceAll | 3.2504 | 2.9970 | **2.8392** | 2.8917 | 2.9055 | 2.8971 |
| FaceFour | 5.6599 | 5.3963 | **5.2482** | 5.3374 | 5.3145 | 5.3444 |
| GunPoint | 2.2964 | **1.6673** | 1.7319 | 1.6998 | 1.7036 | 1.7078 |
| Ham | 4.4250 | 4.1339 | **4.0861** | 4.0997 | 4.0929 | 4.0984 |
| MedicalImages | 2.7714 | **2.6551** | 2.6738 | 2.6693 | 2.6675 | 2.6693 |
| MiddlePhalanxOutlineAgeGroup | 0.5451 | 0.5206 | 0.5138 | 0.5120 | **0.5120** | 0.5143 |
| MiddlePhalanxOutlineCorrect | 0.6671 | 0.6423 | 0.6343 | 0.6341 | 0.6341 | **0.6334** |
| MiddlePhalanxTW | 0.3322 | 0.3284 | 0.3190 | 0.3188 | 0.3189 | **0.3187** |
| MoteStrain | 4.3593 | 4.2874 | 4.2537 | **4.2381** | 4.2390 | 4.2453 |
| ProximalPhalanxTW | 0.3773 | 0.3402 | **0.3365** | 0.3383 | 0.3378 | 0.3382 |
| RefrigerationDevices | 11.5572 | 8.5680 | **7.4193** | 7.9655 | 7.9086 | 7.8243 |
| ScreenType | 13.8207 | 12.2042 | 12.1071 | **12.0787** | 12.0857 | 12.0816 |
| ShapeletSim | 12.2430 | 11.3275 | 11.0399 | 11.0316 | **11.0313** | 11.0834 |
| ShapesAll | 1.2198 | 1.0711 | 1.0687 | **1.0652** | 1.0653 | 1.0690 |
| SmallKitchenAppliances | 13.0036 | 10.3012 | **9.2726** | 9.3408 | 9.3408 | 9.3398 |
| SonyAIBORobotSurface1 | 1.7197 | 1.6757 | 1.6536 | **1.6075** | 1.6088 | 1.6397 |
| SonyAIBORobotSurface2 | 2.7282 | 2.6203 | 2.5849 | **2.5633** | 2.5633 | 2.5633 |
| SyntheticControl | 4.5651 | 4.1545 | 4.1459 | 4.1092 | 4.0827 | **4.0813** |
| Trace | 1.8690 | 0.9323 | 0.8951 | 0.8954 | **0.8499** | 0.8699 |
| TwoLeadECG | 0.9302 | 0.8615 | 0.8463 | 0.8444 | **0.8442** | 0.8473 |
| Wine | 0.3271 | **0.3239** | 0.3257 | 0.3245 | 0.3251 | 0.3240 |
| Worms | 14.1987 | 10.5608 | 10.0867 | 9.9822 | 9.9110 | **9.9007** |
| WormsTwoClass | 9.8468 | 8.7107 | **8.3123** | 8.4709 | 8.4173 | 8.4515 |
| Count | 0 | 3 | 20 | 12 | 6 | 6 |

## B.2 Nearest Centroid Classfication

Table 4: UCR nearest centroid classification accuracy

| Dataset | subgradient | DBA | soft-DTW | ns-DTW ($g_1$) | ns-DTW ($g_2$) | ns-DTW ($g_3$) |
|---|---|---|---|---|---|---|
| Adiac | 0.4490 | 0.4388 | 0.6224 | 0.6327 | **0.65** | 0.6327 |
| ArrowHead | 0.1111 | 0.2222 | **0.33** | 0.2222 | **0.33** | 0.1111 |
| Beef | 0.2500 | 0.2500 | 0.2500 | 0.1250 | **0.38** | 0.2500 |
| BeetleFly | 0.8000 | **1.00** | **1.00** | **1.00** | 0.8000 | 0.8000 |
| BirdChicken | 0.8000 | 0.8000 | 0.8000 | **1.00** | 0.8000 | **1.00** |
| CBF | 0.8750 | **1.00** | **1.00** | **1.00** | 0.8750 | 0.8750 |
| Car | 0.6000 | 0.5333 | **0.73** | 0.6667 | 0.6667 | 0.6667 |
| ChlorineConcentration | 0.2735 | 0.3504 | **0.38** | 0.3675 | 0.3162 | 0.3419 |
| CinCECGTorso | 0.3000 | 0.4000 | 0.4000 | 0.3000 | 0.3000 | **0.50** |
| Coffee | 0.0000 | 0.0000 | 0.0000 | **0.14** | 0.0000 | 0.0000 |
| Computers | 0.6349 | **0.75** | 0.7143 | 0.6667 | 0.6825 | 0.6508 |
| CricketX | 0.3878 | 0.4592 | 0.4694 | 0.5204 | **0.62** | 0.5918 |
| CricketY | 0.3878 | 0.4490 | 0.4694 | 0.4796 | 0.5102 | **0.53** |
| CricketZ | 0.3980 | 0.5408 | 0.5816 | 0.5612 | 0.5510 | **0.65** |
| DistalPhalanxOutlineAgeGroup | 0.8700 | 0.7900 | 0.7900 | 0.8100 | **0.91** | 0.8200 |
| DistalPhalanxOutlineCorrect | 0.1000 | 0.0867 | 0.0933 | 0.1067 | **0.19** | 0.1400 |
| DistalPhalanxTW | **0.04** | 0.0000 | 0.0000 | 0.0000 | 0.0100 | 0.0100 |
| ECG200 | 0.1200 | 0.0800 | 0.0800 | 0.0800 | **0.28** | 0.0000 |
| ECGFiveDays | 0.5000 | 0.5000 | 0.1667 | 0.1667 | **0.67** | **0.67** |
| Earthquakes | 0.0494 | **0.07** | **0.07** | 0.0617 | 0.0370 | 0.0617 |
| FaceAll | 0.8357 | 0.8429 | 0.8714 | 0.9000 | **0.93** | **0.93** |
| FacesUCR | **0.90** | 0.7800 | 0.8600 | 0.8600 | 0.8800 | **0.90** |
| GunPoint | 0.3846 | **0.77** | 0.6923 | 0.6154 | 0.6923 | 0.6154 |
| Ham | 0.6429 | 0.6071 | 0.7143 | **0.79** | 0.6429 | 0.6429 |
| MedicalImages | 0.4583 | 0.4479 | 0.4479 | 0.4375 | 0.4271 | **0.52** |
| MiddlePhalanxOutlineAgeGroup | 0.7600 | **0.81** | **0.81** | **0.81** | 0.7600 | 0.7700 |
| MiddlePhalanxOutlineCorrect | 0.2467 | 0.2067 | 0.2333 | 0.2800 | 0.2267 | **0.39** |
| MiddlePhalanxTW | 0.0300 | 0.0200 | 0.0300 | 0.0100 | 0.0300 | **0.07** |
| MoteStrain | 0.6000 | 0.8000 | 0.8000 | 0.8000 | **1.00** | 0.6000 |
| ProximalPhalanxTW | **0.07** | 0.0100 | 0.0100 | 0.0000 | 0.0000 | 0.0000 |
| RefrigerationDevices | **0.61** | 0.4787 | 0.4787 | 0.5000 | 0.5638 | 0.5106 |
| ScreenType | 0.4255 | 0.4149 | 0.4787 | **0.51** | 0.3936 | 0.3936 |
| ShapeletSim | 0.2000 | 0.4000 | 0.2000 | 0.4000 | **0.60** | 0.2000 |
| ShapesAll | 0.5333 | 0.6600 | 0.6533 | **0.67** | 0.6600 | 0.6067 |
| SmallKitchenAppliances | 0.6277 | 0.6064 | 0.6170 | **0.64** | 0.6064 | 0.5957 |
| SonyAIBORobotSurface1 | **1.00** | 0.6000 | 0.6000 | 0.6000 | **1.00** | 0.6000 |
| SonyAIBORobotSurface2 | 0.8571 | **1.00** | **1.00** | 0.8571 | 0.7143 | 0.5714 |
| SyntheticControl | **1.00** | 0.9867 | **1.00** | 0.9867 | **1.00** | **1.00** |
| Trace | 0.8800 | 0.8800 | 0.9200 | 0.9200 | **1.00** | 0.9600 |
| TwoLeadECG | 0.6667 | 0.8333 | 0.8333 | **1.00** | 0.8333 | 0.8333 |
| Wine | 0.5333 | 0.3333 | 0.3333 | 0.4667 | 0.5333 | **0.67** |
| Worms | 0.4565 | 0.4348 | 0.4783 | **0.50** | 0.3478 | 0.4783 |
| WormsTwoClass | 0.5870 | 0.6739 | 0.6739 | 0.6739 | 0.6522 | **0.70** |
| Count | 6 | 7 | 9 | 11 | 14 | 13 |

## B.3 1NN CLASSIFICATION

Table 5: UCR 1-nearest neighbor classification accuracy

| Dataset | DBA | soft-DTW | ns-DTW ($g_1$) | ns-DTW ($g_2$) | ns-DTW ($g_3$) |
|---|---|---|---|---|---|
| Adiac | 0.5204 | 0.5714 | **0.62** | 0.5714 | 0.5510 |
| ArrowHead | **0.89** | **0.89** | 0.6667 | 0.7778 | **0.89** |
| Beef | 0.2500 | 0.2500 | **0.62** | 0.5000 | 0.2500 |
| BeetleFly | 0.8000 | 0.8000 | 0.8000 | **1.00** | 0.8000 |
| BirdChicken | 0.6000 | 0.6000 | **1.00** | **1.00** | 0.6000 |
| CBF | **1.00** | **1.00** | **1.00** | **1.00** | **1.00** |
| Car | 0.3333 | 0.5333 | **0.67** | 0.4667 | 0.6000 |
| ChlorineConcentration | 0.4359 | 0.4872 | 0.4701 | **0.56** | 0.4957 |
| CinCECGTorso | 0.5000 | 0.5000 | 0.6000 | **0.70** | 0.5000 |
| Coffee | **1.00** | **1.00** | **1.00** | **1.00** | **1.00** |
| Computers | **0.76** | **0.76** | 0.5556 | 0.7460 | **0.76** |
| CricketX | 0.7449 | **0.79** | 0.7245 | 0.7449 | **0.79** |
| CricketY | 0.6735 | 0.6837 | 0.7041 | **0.72** | 0.7041 |
| CricketZ | 0.7347 | 0.7347 | 0.6735 | 0.7143 | **0.74** |
| DistalPhalanxOutlineAgeGroup | 0.7700 | 0.7700 | **0.81** | 0.7800 | 0.7700 |
| DistalPhalanxOutlineCorrect | 0.7733 | 0.7733 | **0.81** | 0.8000 | 0.7667 |
| DistalPhalanxTW | 0.7200 | **0.75** | 0.6900 | 0.7300 | 0.7400 |
| ECG200 | 0.7600 | 0.7600 | 0.7200 | **0.88** | 0.7600 |
| ECGFiveDays | 0.8333 | **1.00** | 0.6667 | 0.5000 | **1.00** |
| Earthquakes | 0.7654 | 0.7654 | **0.78** | 0.7407 | 0.7654 |
| FaceAll | 0.9214 | 0.9500 | 0.9214 | 0.9357 | **0.96** |
| FacesUCR | 0.7600 | **0.78** | 0.6667 | 0.6667 | **0.78** |
| GunPoint | 0.8462 | **0.92** | 0.8600 | 0.9200 | **0.92** |
| Ham | 0.7500 | 0.7500 | **0.92** | 0.7692 | 0.8214 |
| MedicalImages | 0.7083 | 0.7604 | **0.82** | 0.7857 | 0.7083 |
| MiddlePhalanxOutlineAgeGroup | 0.7000 | 0.6900 | 0.6979 | 0.6979 | **0.73** |
| MiddlePhalanxOutlineCorrect | 0.7267 | 0.7733 | **0.85** | 0.8000 | 0.7667 |
| MiddlePhalanxTW | 0.6200 | 0.5900 | 0.7400 | **0.79** | 0.6000 |
| MoteStrain | **0.80** | **0.80** | 0.5900 | 0.5500 | **0.80** |
| ProximalPhalanxTW | 0.7500 | 0.7500 | 0.6000 | **0.80** | 0.7600 |
| RefrigerationDevices | 0.5957 | 0.6383 | 0.7100 | **0.83** | 0.6383 |
| ScreenType | 0.4574 | 0.5106 | **0.70** | 0.6702 | 0.4574 |
| ShapeletSim | 0.2000 | 0.2000 | 0.4787 | **0.52** | 0.2000 |
| ShapesAll | 0.7067 | 0.7467 | 0.6000 | 0.4000 | **0.75** |
| SmallKitchenAppliances | 0.6596 | 0.6915 | 0.7267 | **0.79** | 0.6702 |
| SonyAIBORobotSurface1 | **0.80** | **0.80** | 0.5851 | 0.6383 | **0.80** |
| SonyAIBORobotSurface2 | **0.86** | **0.86** | 0.8000 | 0.8000 | **0.86** |
| SyntheticControl | **1.00** | **1.00** | 0.8571 | 0.7143 | **1.00** |
| Trace | **1.00** | **1.00** | **1.00** | **1.00** | **1.00** |
| TwoLeadECG | **1.00** | **1.00** | 0.9600 | **1.00** | **1.00** |
| Wine | **0.87** | **0.87** | 0.8333 | 0.6667 | **0.87** |
| Worms | 0.4565 | 0.4565 | 0.6667 | **0.87** | 0.4565 |
| WormsTwoClass | 0.7391 | **0.76** | 0.4348 | 0.5000 | 0.7391 |
| Count | 11 | 17 | 14 | 16 | 19 |

## B.4 CLUSTERING

Table 6: UCR Clustering DTW Losses

| Dataset | DBA | soft-DTW | ns-DTW ($g_1$) | ns-DTW ($g_2$) | ns-DTW ($g_3$) |
|---|---|---|---|---|---|
| Adiac | 0.0921 | **0.0878** | 0.0885 | 0.0882 | 0.0885 |
| ArrowHead | 1.2090 | 1.0741 | 1.0624 | **1.0433** | 1.0650 |
| Beef | 2.8558 | 2.2851 | **2.1169** | 2.1680 | 2.1365 |
| BeetleFly | 20.6455 | 19.8514 | 19.0531 | **18.9430** | 19.0679 |
| BirdChicken | 9.4679 | **7.9387** | 8.0163 | 8.0368 | 8.0304 |
| CBF | 12.0459 | 11.8482 | 11.8576 | 11.7534 | **11.7463** |
| Car | 0.8623 | 0.6263 | 0.5699 | **0.5669** | 0.5676 |
| ChlorineConcentration | 6.5869 | 6.4396 | 6.4328 | 6.4346 | **6.4263** |
| CinCECGTorso | **669.7685** | **669.7685** | **669.7685** | **669.7685** | **669.7685** |
| Coffee | 0.4366 | **0.4297** | 0.4305 | 0.4304 | 0.4306 |
| Computers | 143.5164 | 141.2816 | **137.3022** | 141.0881 | 139.3186 |
| CricketX | **114.1204** | **114.1204** | **114.1204** | **114.1204** | **114.1204** |
| CricketY | **103.0182** | **103.0182** | **103.0182** | **103.0182** | **103.0182** |
| CricketZ | **124.6247** | **124.6247** | **124.6247** | **124.6247** | **124.6247** |
| DiatomSizeReduction | 0.0712 | 0.0756 | **0.0628** | 0.0632 | 0.0633 |
| DistalPhalanxOutlineAgeGroup | 0.3678 | **0.3633** | 0.3636 | 0.3637 | 0.3637 |
| DistalPhalanxOutlineCorrect | 0.8450 | **0.7544** | 0.7631 | 0.7635 | 0.7645 |
| DistalPhalanxTW | 0.3325 | 0.3319 | 0.3303 | **0.3292** | 0.3305 |
| ECG200 | 4.5978 | **4.5201** | 4.5334 | 4.5394 | 4.5334 |
| ECG5000 | 6.4048 | 6.6103 | 6.5432 | **6.4040** | 6.5932 |
| ECGFiveDays | 5.7908 | 5.6721 | **5.5425** | 5.5447 | 5.6063 |
| Earthquakes | **437.7200** | **437.7200** | **437.7200** | **437.7200** | **437.7200** |
| FaceAll | **34.8896** | **34.8896** | **34.8896** | **34.8896** | **34.8896** |
| FaceFour | 21.2485 | 20.2836 | 20.1979 | **20.1897** | 20.2069 |
| FacesUCR | **27.0649** | **27.0649** | **27.0649** | **27.0649** | **27.0649** |
| GunPoint | 1.1022 | 1.0099 | 0.9990 | **0.9987** | 0.9989 |
| Ham | 17.1259 | 17.0784 | 17.0572 | **17.0463** | 17.0749 |
| MedicalImages | 4.1345 | 3.5272 | 3.4652 | 3.4744 | **3.4197** |
| MiddlePhalanxOutlineAgeGroup | 0.2368 | **0.2329** | 0.2338 | 0.2335 | 0.2338 |
| MiddlePhalanxOutlineCorrect | 0.3350 | **0.3348** | 0.3348 | 0.3348 | 0.3348 |
| MiddlePhalanxTW | 0.2193 | **0.2134** | 0.2171 | 0.2152 | 0.2172 |
| MoteStrain | 18.8957 | 19.0558 | 19.0490 | 19.0497 | **18.8197** |
| ProximalPhalanxTW | 0.1591 | 0.1584 | 0.1585 | **0.1584** | 0.1585 |
| RefrigerationDevices | 122.9185 | 115.9638 | **113.0523** | 116.9189 | 114.3597 |
| ScreenType | 108.5347 | 104.8087 | 104.5366 | **103.5914** | 104.1625 |
| ShapeletSim | 127.3930 | 122.4791 | 120.5546 | 120.9429 | **120.4151** |
| ShapesAll | **25.5056** | 25.5057 | **25.5056** | **25.5056** | **25.5056** |
| SmallKitchenAppliances | 123.9963 | 123.7793 | **122.3944** | 123.1039 | 123.3957 |
| SonyAIBORobotSurface1 | 4.9519 | 4.7078 | 4.6157 | **4.6127** | 4.6157 |
| SonyAIBORobotSurface2 | 8.9998 | 8.8919 | 8.8860 | 8.8879 | **8.8808** |
| SyntheticControl | 8.7978 | 8.7625 | **8.6648** | 8.6736 | 8.6778 |
| Trace | 3.5626 | 3.4207 | **2.9248** | 2.9365 | 2.9483 |
| TwoLeadECG | 0.9103 | **0.8642** | 0.8644 | 0.8645 | 0.8644 |
| Wine | 0.0697 | **0.0649** | 0.0702 | 0.0691 | 0.0678 |
| WordSynonyms | **27.5408** | **27.5408** | **27.5408** | **27.5408** | **27.5408** |
| Worms | **451.2960** | **451.2960** | **451.2960** | **451.2960** | **451.2960** |
| WormsTwoClass | 81.4757 | 78.6740 | **76.7911** | 78.2881 | 78.3617 |
| Count | 10 | 20 | 19 | 21 | 16 |

# C  ABLATION STUDY OF AVERAGING

Table 7: Ablation Study of Averaging (Grouping $g_1$): Comparison of UCR Barycenter DTW Losses across different parameters $\tau$. Bold indicates the lowest loss.

| Dataset | ns-DTW ($g_1$) | | | |
|---|---|---|---|---|
| | $\tau = 0.95$ | $\tau = 0.90$ | $\tau = 0.85$ | $\tau = 0.80$ |
| Adiac | 0.2551 | 0.2550 | 0.2549 | **0.2549** |
| ArrowHead | 1.1772 | **1.1771** | 1.1772 | 1.1774 |
| Beef | **4.1005** | 4.1014 | 4.1068 | 4.1015 |
| BeetleFly | **4.3015** | 4.3096 | 4.3031 | 4.3037 |
| BirdChicken | 2.4647 | 2.4646 | 2.4649 | **2.4575** |
| CBF | **3.7857** | **3.7857** | **3.7857** | **3.7857** |
| Car | **0.6489** | 0.6493 | 0.6577 | 0.6575 |
| ChlorineConcentration | **3.4750** | **3.4750** | **3.4750** | 3.4750 |
| CinCECGTorso | 8.3677 | 8.5311 | **8.3574** | 8.5775 |
| Coffee | 0.5890 | 0.5890 | 0.5889 | **0.5889** |
| Computers | 14.6314 | 14.6176 | 14.6282 | **14.6000** |
| CricketX | 5.8424 | 5.8500 | **5.8272** | 5.8537 |
| CricketY | **5.5948** | 5.5975 | 5.5974 | 5.5950 |
| CricketZ | 5.3866 | 5.3843 | 5.3875 | **5.3840** |
| DiatomSizeReduction | **0.2278** | 0.2278 | 0.2291 | 0.2289 |
| DistalPhalanxOutlineAgeGroup | **0.7059** | **0.7059** | **0.7059** | **0.7059** |
| DistalPhalanxOutlineCorrect | **0.6666** | **0.6666** | **0.6666** | **0.6666** |
| DistalPhalanxTW | **0.3458** | **0.3458** | **0.3458** | **0.3458** |
| ECG200 | 2.6704 | 2.6704 | 2.6703 | **2.6703** |
| ECG5000 | **2.2057** | 2.2060 | 2.2063 | **2.2057** |
| ECGFiveDays | 2.5275 | **2.4832** | 2.5278 | 2.5324 |
| Earthquakes | **10.8445** | 10.8446 | 10.8446 | 10.8446 |
| FaceAll | **3.0059** | 3.0109 | 3.0110 | 3.0104 |
| FaceFour | **5.3563** | **5.3563** | **5.3563** | **5.3563** |
| FacesUCR | **3.2074** | **3.2074** | **3.2074** | **3.2074** |
| GunPoint | 1.7460 | 1.7440 | **1.7202** | 1.7458 |
| Ham | **4.0997** | 4.1004 | 4.1003 | 4.1019 |
| MedicalImages | 2.6887 | 2.6888 | 2.6887 | **2.6886** |
| MiddlePhalanxOutlineAgeGroup | **0.5152** | **0.5152** | **0.5152** | **0.5152** |
| MiddlePhalanxOutlineCorrect | **0.6389** | **0.6389** | **0.6389** | **0.6389** |
| MiddlePhalanxTW | **0.3282** | 0.3282 | 0.3282 | 0.3282 |
| MoteStrain | 4.2600 | **4.2588** | 4.2589 | 4.2589 |
| ProximalPhalanxTW | **0.3390** | 0.3398 | 0.3396 | 0.3395 |
| RefrigerationDevices | 8.2907 | **8.2763** | 8.3036 | 8.3232 |
| ScreenType | 12.0787 | 12.0795 | 12.1121 | **12.0785** |
| ShapeletSim | 11.0850 | **11.0820** | 11.1077 | 11.1077 |
| ShapesAll | 1.0652 | 1.0652 | **1.0638** | 1.0657 |
| SmallKitchenAppliances | 9.9110 | 9.9131 | **9.3478** | 9.3585 |
| SonyAIBORobotSurface1 | **1.6774** | **1.6774** | **1.6774** | **1.6774** |
| SonyAIBORobotSurface2 | **2.5914** | **2.5914** | **2.5914** | **2.5914** |
| SyntheticControl | **4.1615** | **4.1615** | **4.1615** | **4.1615** |
| Trace | 0.8954 | **0.8949** | 0.8969 | 0.8962 |
| TwoLeadECG | 0.8555 | 0.8554 | 0.8554 | **0.8553** |
| Wine | **0.3245** | 0.3245 | 0.3245 | 0.3245 |
| WordSynonyms | 2.1832 | **2.1731** | 2.1836 | 2.1744 |
| Worms | **10.2557** | 10.4207 | 10.4068 | 10.4219 |
| WormsTwoClass | 8.4709 | **8.4596** | 8.4805 | 8.4603 |
| Count | 25 | 20 | 17 | 21 |

Table 8: Ablation Study of Averaging (Grouping $g_2$): Comparison of UCR Barycenter DTW Losses across different parameters $\tau$. Bold indicates the lowest loss.

| Dataset | ns-DTW ($g_2$) | | | |
|---|---|---|---|---|
| | $\tau = 0.95$ | $\tau = 0.90$ | $\tau = 0.85$ | $\tau = 0.80$ |
| Adiac | 0.2553 | 0.2553 | 0.2553 | **0.2552** |
| ArrowHead | 1.1663 | **1.1651** | 1.1652 | 1.1846 |
| Beef | 4.1025 | 4.1114 | 4.1036 | **4.0924** |
| BeetleFly | 4.3052 | 4.3067 | 4.3170 | **4.3045** |
| BirdChicken | 2.4589 | 2.4597 | **2.4540** | 2.4633 |
| CBF | 3.8330 | **3.8132** | 3.8132 | 3.8140 |
| Car | **0.6492** | 0.6493 | 0.6571 | 0.6576 |
| ChlorineConcentration | 3.4750 | 3.4750 | **3.4749** | 3.4750 |
| CinCECGTorso | **8.3597** | 8.4085 | 8.5551 | 8.4076 |
| Coffee | **0.5892** | 0.5894 | 0.5906 | 0.5901 |
| Computers | 14.6228 | 14.6119 | 14.6227 | **14.6075** |
| CricketX | 5.8423 | **5.8244** | 5.8738 | 5.8396 |
| CricketY | **5.5944** | 5.5968 | 5.5980 | 5.5974 |
| CricketZ | 5.3852 | 5.3829 | 5.3826 | **5.3826** |
| DiatomSizeReduction | 0.2278 | 0.2275 | 0.2274 | **0.2271** |
| DistalPhalanxOutlineAgeGroup | 0.7059 | **0.7058** | 0.7059 | 0.7058 |
| DistalPhalanxOutlineCorrect | **0.6666** | **0.6666** | **0.6666** | **0.6666** |
| DistalPhalanxTW | **0.3458** | 0.3458 | 0.3458 | 0.3459 |
| ECG200 | 2.6726 | **2.6718** | 2.6738 | 2.6720 |
| ECG5000 | 2.2080 | 2.2074 | 2.2065 | **2.2011** |
| ECGFiveDays | **2.5185** | 2.5265 | 2.5293 | 2.5237 |
| Earthquakes | **10.8293** | 10.8420 | 10.8365 | 10.8365 |
| FaceAll | 3.0224 | 3.0080 | **3.0065** | 3.0178 |
| FaceFour | **5.3563** | 5.3564 | 5.3564 | 5.3565 |
| GunPoint | 1.7464 | **1.7436** | 1.7441 | 1.7451 |
| Ham | 4.0929 | **4.0802** | 4.1005 | 4.1018 |
| MedicalImages | **2.6858** | 2.6864 | 2.6860 | 2.6870 |
| MiddlePhalanxOutlineAgeGroup | 0.5145 | **0.5142** | 0.5143 | 0.5143 |
| MiddlePhalanxOutlineCorrect | 0.6389 | 0.6389 | **0.6388** | 0.6388 |
| MiddlePhalanxTW | **0.3282** | 0.3282 | 0.3282 | 0.3282 |
| MoteStrain | 4.2591 | 4.2565 | **4.2546** | 4.2567 |
| ProximalPhalanxTW | 0.3399 | **0.3397** | 0.3399 | 0.3399 |
| RefrigerationDevices | 8.9489 | 8.9519 | 8.9514 | **8.8522** |
| ScreenType | 12.0857 | 12.0816 | **12.0815** | 12.0838 |
| ShapeletSim | **11.0833** | 11.0836 | 11.1077 | 11.1078 |
| ShapesAll | **1.0653** | 1.0740 | 1.0662 | 1.0680 |
| SmallKitchenAppliances | 9.4743 | **9.4160** | 9.4293 | 9.9123 |
| SonyAIBORobotSurface1 | **1.6774** | **1.6774** | **1.6774** | **1.6774** |
| SonyAIBORobotSurface2 | **2.5914** | **2.5914** | **2.5914** | **2.5914** |
| SyntheticControl | 4.1615 | 4.1615 | 4.1615 | **4.1615** |
| Trace | **0.8949** | 0.9133 | 0.8964 | 0.9030 |
| TwoLeadECG | 0.8554 | 0.8549 | 0.8551 | **0.8547** |
| Wine | **0.3251** | 0.3253 | 0.3264 | 0.3260 |
| Worms | 10.4246 | **10.4022** | 10.4252 | 10.4123 |
| WormsTwoClass | 8.5204 | **8.4643** | 8.4992 | 8.5186 |
| Count | 17 | 15 | 10 | 13 |

Table 9: Ablation Study of Averaging (Grouping $g_3$): Comparison of UCR Barycenter DTW Losses across different parameters $\tau$. Bold indicates the lowest loss.

| Dataset | ns-DTW ($g_3$) | | | |
|---|---|---|---|---|
| | $\tau = 0.95$ | $\tau = 0.90$ | $\tau = 0.85$ | $\tau = 0.80$ |
| Adiac | **0.2552** | 0.2553 | 0.2554 | 0.2556 |
| ArrowHead | **1.1643** | 1.1774 | 1.1789 | 1.1775 |
| Beef | **4.1015** | 4.1018 | 4.1054 | 4.1238 |
| BeetleFly | **4.3045** | 4.3085 | 4.3055 | 4.3070 |
| BirdChicken | 2.4642 | 2.4654 | 2.4548 | **2.4548** |
| CBF | 3.8332 | 3.8331 | **3.8132** | 3.8140 |
| Car | 0.6489 | 0.6486 | **0.6446** | 0.6485 |
| ChlorineConcentration | 3.4751 | 3.4746 | **3.4745** | 3.4746 |
| CinCECGTorso | **8.3599** | 8.5016 | 8.3633 | 8.5583 |
| Coffee | **0.5893** | 0.5903 | 0.5904 | 0.5905 |
| Computers | 14.6203 | **14.6059** | 14.6578 | 14.6558 |
| CricketX | 5.8534 | 5.9529 | **5.8229** | 6.0129 |
| CricketY | 5.5951 | 5.5954 | **5.5947** | 5.5952 |
| CricketZ | 5.3862 | 5.3891 | 5.3829 | **5.3817** |
| DiatomSizeReduction | **0.2278** | 0.2280 | 0.2285 | 0.2279 |
| DistalPhalanxOutlineAgeGroup | 0.7057 | 0.7060 | 0.7055 | **0.7047** |
| DistalPhalanxOutlineCorrect | **0.6666** | **0.6666** | **0.6666** | **0.6666** |
| DistalPhalanxTW | 0.3458 | 0.3459 | 0.3458 | **0.3458** |
| ECG200 | 2.6704 | 2.6704 | **2.6704** | **2.6704** |
| ECG5000 | 2.2060 | 2.2061 | 2.2056 | **2.2000** |
| ECGFiveDays | 2.5252 | 2.5248 | 2.5262 | **2.4842** |
| Earthquakes | **10.8283** | 10.8446 | 10.8447 | 10.8446 |
| FaceAll | 3.0104 | 3.0102 | 3.0108 | **3.0096** |
| FaceFour | **5.3562** | 5.3562 | 5.3562 | 5.3562 |
| FacesUCR | **3.2074** | **3.2074** | **3.2074** | **3.2074** |
| GunPoint | **1.7408** | 1.7427 | 1.7440 | 1.7423 |
| Ham | **4.0984** | 4.0993 | 4.0993 | 4.1070 |
| MedicalImages | 2.6887 | 2.6896 | 2.6896 | **2.6886** |
| MiddlePhalanxOutlineAgeGroup | 0.5143 | 0.5141 | **0.5141** | **0.5141** |
| MiddlePhalanxOutlineCorrect | **0.6389** | 0.6390 | 0.6390 | 0.6390 |
| MiddlePhalanxTW | **0.3281** | **0.3281** | **0.3281** | **0.3281** |
| MoteStrain | 4.2582 | **4.2565** | 4.2572 | 4.2571 |
| ProximalPhalanxTW | 0.3399 | 0.3400 | 0.3398 | **0.3397** |
| RefrigerationDevices | **8.8975** | 8.9557 | 8.9514 | 8.9532 |
| ScreenType | **12.0816** | 12.0858 | 12.0848 | 12.0858 |
| ShapeletSim | **11.0834** | 11.1077 | 11.1077 | 11.1077 |
| ShapesAll | 1.0690 | 1.0688 | 1.0683 | **1.0641** |
| SmallKitchenAppliances | 9.4851 | **9.3029** | 9.7840 | 9.7895 |
| SonyAIBORobotSurface1 | 1.6774 | 1.6773 | **1.6773** | **1.6773** |
| SonyAIBORobotSurface2 | **2.5914** | **2.5914** | 2.5914 | 2.5914 |
| SyntheticControl | **4.1615** | **4.1615** | **4.1615** | **4.1615** |
| Trace | 0.8970 | 0.8961 | 0.9118 | **0.8960** |
| TwoLeadECG | 0.8554 | 0.8554 | 0.8553 | **0.8553** |
| Wine | 0.3240 | **0.3238** | 0.3242 | 0.3240 |
| WordSynonyms | 2.1858 | **2.1513** | 2.1593 | 2.1863 |
| Worms | 10.4158 | **10.2059** | 10.4138 | 10.4293 |
| WormsTwoClass | 8.4515 | 8.4789 | 8.4532 | **8.4484** |
| Count | 20 | 11 | 12 | 20 |

# D ABLATION STUDY OF CLASSIFICATION

Table 10: Ablation Study of Classification (Grouping $g_1$): Comparison of nearest centroid classification accuracy across different parameters $\tau$. Bold indicates the highest accuracy.

| Dataset | ns-DTW ($g_1$) | | | |
| --- | --- | --- | --- | --- |
| | $\tau = 0.95$ | $\tau = 0.90$ | $\tau = 0.85$ | $\tau = 0.80$ |
| Adiac | **0.63** | 0.5306 | 0.5918 | 0.5408 |
| ArrowHead | 0.2222 | 0.2222 | **0.33** | 0.2222 |
| Beef | 0.1250 | **0.50** | 0.2500 | 0.3750 |
| BeetleFly | **1.00** | 0.8000 | 0.8000 | 0.8000 |
| BirdChicken | **1.00** | 0.6000 | 0.6000 | 0.6000 |
| CBF | **1.00** | **1.00** | 0.8750 | **1.00** |
| Car | 0.6667 | 0.7333 | 0.7333 | **0.80** |
| ChlorineConcentration | **0.37** | 0.3162 | 0.2137 | 0.3504 |
| CinCECGTorso | 0.3000 | 0.5000 | **0.60** | 0.3000 |
| Coffee | **0.14** | 0.0000 | 0.0000 | **0.14** |
| Computers | 0.6667 | 0.5397 | **0.68** | 0.5714 |
| CricketX | 0.5204 | 0.6122 | 0.5000 | **0.63** |
| CricketY | 0.4796 | 0.5612 | 0.4490 | **0.57** |
| CricketZ | 0.5612 | 0.5816 | 0.5408 | **0.59** |
| DistalPhalanxOutlineAgeGroup | 0.8100 | 0.8400 | 0.8300 | **0.85** |
| DistalPhalanxOutlineCorrect | 0.1067 | 0.2200 | **0.23** | 0.0867 |
| DistalPhalanxTW | 0.0000 | 0.0200 | 0.0300 | **0.06** |
| ECG200 | 0.0800 | 0.1600 | 0.1600 | **0.28** |
| ECGFiveDays | 0.1667 | **0.67** | 0.5000 | 0.5000 |
| Earthquakes | **0.06** | 0.0123 | 0.0123 | **0.06** |
| FaceAll | 0.9000 | **0.92** | 0.9143 | 0.9143 |
| FacesUCR | **0.86** | **0.86** | **0.86** | 0.8000 |
| GunPoint | 0.6154 | 0.6154 | 0.6154 | **0.85** |
| Ham | **0.79** | 0.6429 | 0.7500 | 0.6429 |
| MedicalImages | 0.4375 | 0.4271 | **0.47** | **0.47** |
| MiddlePhalanxOutlineAgeGroup | **0.81** | 0.7300 | 0.7700 | 0.8000 |
| MiddlePhalanxOutlineCorrect | **0.28** | 0.2533 | 0.2267 | 0.2067 |
| MiddlePhalanxTW | 0.0100 | **0.05** | **0.05** | **0.05** |
| MoteStrain | **0.80** | **0.80** | **0.80** | **0.80** |
| ProximalPhalanxTW | **0.00** | **0.00** | **0.00** | **0.00** |
| RefrigerationDevices | 0.5000 | 0.5426 | **0.60** | 0.5106 |
| ScreenType | **0.51** | 0.4149 | 0.4043 | 0.4574 |
| ShapeletSim | **0.40** | 0.0000 | 0.2000 | **0.40** |
| ShapesAll | **0.67** | 0.6133 | 0.6600 | 0.6600 |
| SmallKitchenAppliances | **0.64** | 0.5745 | 0.5745 | **0.64** |
| SonyAIBORobotSurface1 | 0.6000 | **1.00** | 0.8000 | **1.00** |
| SonyAIBORobotSurface2 | 0.8571 | 0.8571 | 0.5714 | **1.00** |
| SyntheticControl | 0.9867 | **1.00** | **1.00** | **1.00** |
| Trace | 0.9200 | **1.00** | 0.8400 | **1.00** |
| TwoLeadECG | **1.00** | 0.6667 | 0.6667 | 0.8333 |
| Wine | 0.4667 | 0.3333 | **0.73** | 0.4667 |
| Worms | **0.50** | 0.2826 | 0.3913 | 0.4130 |
| WormsTwoClass | **0.67** | 0.5652 | 0.5435 | 0.5652 |
| Count | 20 | 11 | 12 | 21 |

Table 11: Ablation Study of Classification (Grouping $g_2$): Comparison of nearest centroid classification accuracy across different parameters $\tau$. Bold indicates the highest accuracy.

| Dataset | ns-DTW ($g_2$) | | | |
|---|---|---|---|---|
| | $\tau = 0.95$ | $\tau = 0.90$ | $\tau = 0.85$ | $\tau = 0.80$ |
| Adiac | **0.65** | 0.5510 | 0.6122 | 0.5714 |
| ArrowHead | **0.33** | 0.1111 | 0.0000 | 0.1111 |
| Beef | 0.3750 | **0.50** | **0.50** | 0.3750 |
| BeetleFly | **0.80** | **0.80** | 0.6000 | 0.6000 |
| BirdChicken | 0.8000 | **1.00** | 0.8000 | 0.8000 |
| CBF | **0.88** | **0.88** | **0.88** | **0.88** |
| Car | **0.67** | **0.67** | **0.67** | **0.67** |
| ChlorineConcentration | 0.3162 | 0.2906 | 0.3675 | **0.38** |
| CinCECGTorso | 0.3000 | 0.3000 | 0.4000 | **0.50** |
| Coffee | **0.00** | **0.00** | **0.00** | **0.00** |
| Computers | **0.68** | 0.6508 | 0.6508 | 0.6349 |
| CricketX | 0.6224 | 0.5204 | 0.5714 | **0.65** |
| CricketY | 0.5102 | **0.61** | 0.5612 | 0.5918 |
| CricketZ | 0.5510 | **0.61** | 0.4592 | 0.4796 |
| DistalPhalanxOutlineAgeGroup | **0.91** | 0.8500 | 0.8800 | 0.8400 |
| DistalPhalanxOutlineCorrect | 0.1867 | 0.1400 | 0.1667 | **0.29** |
| DistalPhalanxTW | 0.0100 | 0.0100 | 0.0000 | **0.02** |
| ECG200 | **0.28** | 0.2400 | 0.2000 | 0.1200 |
| ECGFiveDays | 0.6667 | 0.5000 | 0.6667 | **0.83** |
| Earthquakes | 0.0370 | 0.0247 | **0.05** | 0.0370 |
| FaceAll | **0.93** | 0.9143 | 0.8643 | 0.9000 |
| FacesUCR | 0.8800 | **0.94** | 0.8200 | 0.9000 |
| GunPoint | **0.69** | **0.69** | 0.6154 | **0.69** |
| Ham | 0.6429 | 0.6071 | 0.7500 | **0.86** |
| MedicalImages | 0.4271 | 0.4271 | 0.3646 | **0.49** |
| MiddlePhalanxOutlineAgeGroup | 0.7600 | **0.78** | 0.7500 | 0.7500 |
| MiddlePhalanxOutlineCorrect | 0.2267 | 0.2600 | 0.3600 | **0.41** |
| MiddlePhalanxTW | 0.0300 | 0.0500 | **0.07** | 0.0600 |
| MoteStrain | **1.00** | 0.8000 | 0.8000 | 0.6000 |
| ProximalPhalanxTW | 0.0000 | **0.01** | 0.0000 | 0.0000 |
| RefrigerationDevices | 0.5638 | **0.68** | 0.5426 | 0.5426 |
| ScreenType | 0.3936 | **0.48** | 0.4149 | 0.4255 |
| ShapeletSim | **0.60** | 0.2000 | 0.2000 | 0.4000 |
| ShapesAll | 0.6600 | 0.6533 | 0.6467 | **0.68** |
| SmallKitchenAppliances | 0.6064 | **0.73** | 0.6383 | 0.5957 |
| SonyAIBORobotSurface1 | **1.00** | 0.8000 | **1.00** | **1.00** |
| SonyAIBORobotSurface2 | 0.7143 | 0.7143 | **1.00** | **1.00** |
| SyntheticControl | **1.00** | 0.9867 | **1.00** | 0.9867 |
| Trace | **1.00** | **1.00** | 0.9600 | **1.00** |
| TwoLeadECG | **0.83** | 0.6667 | 0.6667 | 0.6667 |
| Wine | **0.53** | **0.53** | **0.53** | 0.2667 |
| Worms | 0.3478 | 0.5000 | 0.3696 | **0.52** |
| WormsTwoClass | 0.6522 | 0.5217 | **0.67** | 0.6087 |
| Count | 18 | 17 | 11 | 18 |

Table 12: Ablation Study of Classification (Grouping $g_3$): Comparison of nearest centroid classification accuracy across different parameters $\tau$. Bold indicates the highest accuracy.

| Dataset | ns-DTW ($g_3$) | | | |
| --- | --- | --- | --- | --- |
| | $\tau = 0.95$ | $\tau = 0.90$ | $\tau = 0.85$ | $\tau = 0.80$ |
| Adiac | **0.63** | 0.6020 | 0.5918 | 0.6020 |
| ArrowHead | 0.1111 | **0.33** | **0.33** | 0.2222 |
| Beef | **0.25** | 0.1250 | 0.1250 | **0.25** |
| BeetleFly | **0.80** | 0.6000 | 0.6000 | **0.80** |
| BirdChicken | **1.00** | 0.6000 | 0.4000 | 0.8000 |
| CBF | 0.8750 | **1.00** | 0.8750 | 0.8750 |
| Car | 0.6667 | **0.80** | 0.7333 | **0.80** |
| ChlorineConcentration | **0.34** | 0.2564 | 0.3077 | 0.3333 |
| CinCECGTorso | **0.50** | **0.50** | 0.4000 | 0.2000 |
| Coffee | 0.0000 | 0.0000 | 0.0000 | **0.14** |
| Computers | **0.65** | **0.65** | 0.5873 | 0.6190 |
| CricketX | **0.59** | 0.5612 | 0.5612 | 0.5510 |
| CricketY | 0.5306 | 0.5102 | **0.56** | 0.5408 |
| CricketZ | **0.65** | 0.5306 | 0.5000 | 0.5204 |
| DistalPhalanxOutlineAgeGroup | 0.8200 | **0.86** | 0.8100 | 0.8500 |
| DistalPhalanxOutlineCorrect | 0.1400 | **0.26** | 0.0867 | 0.1933 |
| DistalPhalanxTW | 0.0100 | **0.03** | 0.0100 | 0.0200 |
| ECG200 | 0.0000 | **0.20** | 0.1200 | 0.1200 |
| ECGFiveDays | 0.6667 | 0.5000 | **1.00** | 0.6667 |
| Earthquakes | **0.06** | 0.0247 | 0.0247 | 0.0247 |
| FaceAll | **0.93** | 0.9143 | 0.9143 | 0.8857 |
| FacesUCR | 0.9000 | 0.7600 | 0.9200 | **0.94** |
| GunPoint | 0.6154 | **0.77** | **0.77** | 0.4615 |
| Ham | 0.6429 | 0.7143 | **0.75** | **0.75** |
| MedicalImages | **0.52** | 0.4271 | 0.3542 | 0.3542 |
| MiddlePhalanxOutlineAgeGroup | 0.7700 | 0.7500 | **0.80** | 0.7900 |
| MiddlePhalanxOutlineCorrect | 0.3867 | 0.3667 | **0.43** | 0.2867 |
| MiddlePhalanxTW | **0.07** | 0.0600 | 0.0400 | 0.0400 |
| MoteStrain | 0.6000 | **0.80** | 0.6000 | **0.80** |
| ProximalPhalanxTW | **0.00** | **0.00** | **0.00** | **0.00** |
| RefrigerationDevices | 0.5106 | 0.5426 | **0.61** | 0.5319 |
| ScreenType | 0.3936 | **0.44** | 0.4255 | 0.4149 |
| ShapeletSim | 0.2000 | **0.40** | 0.2000 | **0.40** |
| ShapesAll | 0.6067 | 0.6467 | **0.72** | 0.6533 |
| SmallKitchenAppliances | 0.5957 | **0.66** | 0.6489 | 0.6277 |
| SonyAIBORobotSurface1 | 0.6000 | 0.8000 | **1.00** | 0.8000 |
| SonyAIBORobotSurface2 | 0.5714 | **1.00** | **1.00** | 0.7143 |
| SyntheticControl | **1.00** | **1.00** | 0.9867 | 0.9867 |
| Trace | **0.96** | **0.96** | 0.9200 | **0.96** |
| TwoLeadECG | 0.8333 | 0.8333 | 0.3333 | **1.00** |
| Wine | **0.67** | 0.5333 | 0.4667 | 0.4000 |
| Worms | **0.48** | 0.4348 | 0.4130 | 0.3696 |
| WormsTwoClass | **0.70** | 0.6522 | 0.6304 | 0.5870 |
| Count | 19 | 18 | 12 | 11 |

# E ABLATION STUDY FOR CLUSTERING

Table 13: Ablation Study of Clustering (Grouping $g_1$): Comparison of UCR Clustering DTW Losses across different parameters $\tau$. Bold indicates the lowest loss.

| Dataset | ns-DTW ($g_1$) | | | |
|---|---|---|---|---|
| | $\tau = 0.95$ | $\tau = 0.90$ | $\tau = 0.85$ | $\tau = 0.80$ |
| Adiac | 0.0933 | 0.0887 | 0.0887 | **0.0885** |
| ArrowHead | **1.0624** | 1.0781 | 1.0750 | 1.0723 |
| Beef | 2.6543 | 2.3194 | **2.1169** | 2.6488 |
| BeetleFly | **19.0531** | 19.0865 | 19.0850 | 19.1989 |
| BirdChicken | **8.0163** | 8.1141 | 8.0221 | 8.0705 |
| CBF | 11.9793 | 11.8627 | **11.8576** | 11.8919 |
| Car | 0.5828 | 0.6079 | **0.5699** | 0.5820 |
| ChlorineConcentration | **6.4328** | 6.4504 | 6.4373 | 6.4452 |
| CinCECGTorso | **669.7685** | **669.7685** | **669.7685** | **669.7685** |
| Coffee | 0.4340 | 0.4313 | 0.4305 | **0.4305** |
| Computers | 141.2125 | 142.1296 | 142.4221 | **137.3022** |
| CricketX | **114.1204** | **114.1204** | **114.1204** | **114.1204** |
| CricketY | **103.0182** | **103.0182** | **103.0182** | **103.0182** |
| CricketZ | **124.6247** | **124.6247** | **124.6247** | **124.6247** |
| DiatomSizeReduction | 0.1008 | 0.0634 | 0.0631 | **0.0628** |
| DistalPhalanxOutlineAgeGroup | **0.3636** | 0.3646 | 0.3639 | 0.3639 |
| DistalPhalanxOutlineCorrect | 0.7662 | **0.7631** | 0.7662 | 0.7644 |
| DistalPhalanxTW | 0.3305 | 0.3318 | 0.3305 | **0.3303** |
| ECG200 | 4.5388 | 4.5473 | **4.5334** | 4.5376 |
| ECG5000 | 6.7068 | 6.6826 | 6.5888 | **6.5432** |
| ECGFiveDays | 5.7248 | **5.5425** | 5.5456 | 5.6963 |
| Earthquakes | **437.7200** | **437.7200** | **437.7200** | **437.7200** |
| FaceAll | 34.8896 | **34.8896** | **34.8896** | **34.8896** |
| FaceFour | 20.2089 | 20.2031 | **20.1979** | 20.3103 |
| FacesUCR | **27.0649** | **27.0649** | **27.0649** | **27.0649** |
| GunPoint | 1.0274 | **0.9990** | 1.0012 | 1.0023 |
| Ham | 17.0643 | 17.0764 | 17.0641 | **17.0572** |
| MedicalImages | **3.4652** | 3.5714 | 3.5776 | 3.6044 |
| MiddlePhalanxOutlineAgeGroup | **0.2338** | 0.2349 | 0.2342 | 0.2340 |
| MiddlePhalanxOutlineCorrect | 0.3353 | 0.3349 | 0.3349 | **0.3348** |
| MiddlePhalanxTW | **0.2171** | 0.2180 | 0.2171 | 0.2178 |
| MoteStrain | **19.0490** | 19.0853 | 19.0503 | 19.0503 |
| ProximalPhalanxTW | 0.1595 | 0.1585 | **0.1585** | 0.1585 |
| RefrigerationDevices | **113.0523** | 114.4612 | 114.9686 | 117.9007 |
| ScreenType | **104.5366** | 105.8296 | 105.6790 | 104.8913 |
| ShapeletSim | 121.8188 | 120.5682 | **120.5546** | 120.7565 |
| ShapesAll | 25.5065 | **25.5056** | **25.5056** | **25.5056** |
| SmallKitchenAppliances | 124.1084 | **122.3944** | 123.6107 | 123.2592 |
| SonyAIBORobotSurface1 | **4.6157** | 4.6442 | 4.6401 | 4.6435 |
| SonyAIBORobotSurface2 | 8.8879 | 8.8915 | 8.8879 | **8.8860** |
| SyntheticControl | **8.6648** | 8.6900 | 8.6810 | 8.6787 |
| Trace | 2.9443 | 2.9413 | 2.9340 | **2.9248** |
| TwoLeadECG | 0.8646 | 0.8769 | 0.8648 | **0.8644** |
| Wine | 0.0704 | 0.0702 | **0.0702** | 0.0702 |
| WordSynonyms | **27.5408** | **27.5408** | **27.5408** | **27.5408** |
| Worms | **451.2960** | **451.2960** | **451.2960** | **451.2960** |
| WormsTwoClass | 78.2919 | 78.5687 | **76.7911** | 78.7067 |
| Count | 21 | 14 | 19 | 21 |

Table 14: Ablation Study of Clustering (Grouping $g_2$): Comparison of UCR Clustering DTW Losses across different parameters $\tau$. Bold indicates the lowest loss.

| Dataset | ns-DTW ($g_2$) | | | |
|---|---|---|---|---|
| | $\tau = 0.95$ | $\tau = 0.90$ | $\tau = 0.85$ | $\tau = 0.80$ |
| Adiac | 0.0889 | 0.0886 | **0.0882** | 0.0887 |
| ArrowHead | 1.0626 | 1.0754 | 1.0447 | **1.0433** |
| Beef | 2.6517 | 2.6512 | 2.2552 | **2.1680** |
| BeetleFly | 19.1110 | 19.1173 | **18.9430** | 19.0810 |
| BirdChicken | 8.0838 | **8.0368** | 8.0901 | 8.0716 |
| CBF | 11.8548 | 11.8228 | **11.7534** | 11.8266 |
| Car | **0.5669** | 0.5728 | 0.5735 | 0.5785 |
| ChlorineConcentration | **6.4346** | 6.4438 | 6.4461 | 6.4400 |
| CinCECGTorso | **669.7685** | **669.7685** | **669.7685** | **669.7685** |
| Coffee | 0.4305 | **0.4304** | 0.4307 | 0.4314 |
| Computers | 143.9456 | **141.0881** | 142.3675 | 141.7750 |
| CricketX | **114.1204** | **114.1204** | **114.1204** | **114.1204** |
| CricketY | **103.0182** | **103.0182** | **103.0182** | **103.0182** |
| CricketZ | **124.6247** | **124.6247** | **124.6247** | **124.6247** |
| DiatomSizeReduction | **0.0632** | 0.0635 | 0.0633 | 0.0636 |
| DistalPhalanxOutlineAgeGroup | 0.3641 | 0.3640 | **0.3637** | 0.3639 |
| DistalPhalanxOutlineCorrect | 0.7654 | 0.7648 | 0.7641 | **0.7635** |
| DistalPhalanxTW | 0.3305 | 0.3305 | 0.3307 | **0.3292** |
| ECG200 | **4.5394** | 4.5605 | 4.5595 | 4.5452 |
| ECG5000 | 6.7502 | 6.5538 | **6.4040** | 6.5952 |
| ECGFiveDays | 5.6538 | 5.6400 | 5.5714 | **5.5447** |
| Earthquakes | **437.7200** | **437.7200** | **437.7200** | **437.7200** |
| FaceAll | **34.8896** | **34.8896** | **34.8896** | **34.8896** |
| FaceFour | 20.2089 | 20.2024 | 20.1946 | **20.1897** |
| FacesUCR | **27.0649** | **27.0649** | **27.0649** | **27.0649** |
| GunPoint | **0.9987** | 1.0007 | 1.0040 | 1.0036 |
| Ham | 17.0729 | 17.0593 | **17.0463** | 17.1152 |
| MedicalImages | **3.4744** | 3.5939 | 3.5652 | 3.5336 |
| MiddlePhalanxOutlineAgeGroup | 0.2336 | **0.2335** | 0.2337 | 0.2336 |
| MiddlePhalanxOutlineCorrect | 0.3349 | 0.3348 | **0.3348** | 0.3349 |
| MiddlePhalanxTW | **0.2152** | 0.2169 | 0.2169 | 0.2173 |
| MoteStrain | **19.0497** | 19.0506 | 19.0507 | 19.0507 |
| ProximalPhalanxTW | **0.1584** | 0.1584 | 0.1585 | 0.1586 |
| RefrigerationDevices | 119.7300 | **116.9189** | 119.4748 | 119.4159 |
| ScreenType | 104.6065 | 104.5283 | **103.5914** | 104.3206 |
| ShapeletSim | 121.4127 | 121.6850 | 122.1467 | **120.9429** |
| ShapesAll | **25.5056** | **25.5056** | **25.5056** | **25.5056** |
| SmallKitchenAppliances | **123.1039** | 123.6502 | 123.9712 | 124.0442 |
| SonyAIBORobotSurface1 | **4.6127** | 4.6506 | 4.6475 | 4.6339 |
| SonyAIBORobotSurface2 | **8.8879** | 8.8879 | 8.8879 | 8.8879 |
| SyntheticControl | 8.6874 | 8.6919 | 8.6829 | **8.6736** |
| Trace | 2.9486 | 2.9450 | 2.9467 | **2.9365** |
| TwoLeadECG | 0.8646 | **0.8645** | 0.8646 | 0.8652 |
| Wine | 0.0699 | 0.0694 | 0.0694 | **0.0691** |
| WordSynonyms | **27.5408** | **27.5408** | **27.5408** | **27.5408** |
| Worms | **451.2960** | **451.2960** | **451.2960** | **451.2960** |
| WormsTwoClass | 78.6962 | **78.2881** | 78.6430 | 78.4662 |
| Count | 22 | 17 | 18 | 20 |

Table 15: Ablation Study of Clustering (Grouping $g_3$): Comparison of UCR Clustering DTW Losses across different parameters $\tau$. Bold indicates the lowest loss.

| Dataset | ns-DTW ($g_3$) | | | |
|---|---|---|---|---|
| | $\tau = 0.95$ | $\tau = 0.90$ | $\tau = 0.85$ | $\tau = 0.80$ |
| Adiac | **0.0885** | 0.0888 | 0.0888 | 0.0885 |
| ArrowHead | **1.0650** | 1.0663 | 1.0664 | 1.0665 |
| Beef | 2.2629 | 2.2551 | 2.2545 | **2.1365** |
| BeetleFly | 19.0952 | 19.0988 | 19.0856 | **19.0679** |
| BirdChicken | 8.0704 | 8.0366 | **8.0304** | 8.1060 |
| CBF | 11.7832 | **11.7463** | 11.8067 | 11.8216 |
| Car | **0.5676** | 0.5790 | 0.5970 | 0.5846 |
| ChlorineConcentration | **6.4263** | 6.4330 | 6.4319 | 6.4301 |
| CinCECGTorso | **669.7685** | 669.7685 | 669.7685 | 669.7685 |
| Coffee | **0.4306** | 0.4306 | 0.4307 | 0.4318 |
| Computers | 139.4716 | 140.3803 | **139.3186** | 142.6158 |
| CricketX | **114.1204** | **114.1204** | **114.1204** | **114.1204** |
| CricketY | **103.0182** | **103.0182** | **103.0182** | **103.0182** |
| CricketZ | **124.6247** | **124.6247** | **124.6247** | **124.6247** |
| DiatomSizeReduction | 0.0634 | 0.0634 | **0.0633** | 0.0635 |
| DistalPhalanxOutlineAgeGroup | 0.3640 | 0.3638 | 0.3640 | **0.3637** |
| DistalPhalanxOutlineCorrect | 0.7659 | **0.7645** | 0.7688 | 0.7692 |
| DistalPhalanxTW | **0.3305** | 0.3306 | 0.3318 | 0.3317 |
| ECG200 | 4.5356 | 4.5382 | **4.5334** | 4.5380 |
| ECG5000 | 6.6823 | **6.5932** | 6.7333 | 6.6061 |
| ECGFiveDays | **5.6063** | 5.6822 | 5.6325 | 5.7113 |
| Earthquakes | **437.7200** | **437.7200** | **437.7200** | **437.7200** |
| FaceAll | **34.8896** | **34.8896** | **34.8896** | **34.8896** |
| FaceFour | 20.2126 | **20.2069** | 20.2224 | 20.3481 |
| FacesUCR | **27.0649** | **27.0649** | **27.0649** | **27.0649** |
| GunPoint | 1.0012 | **0.9989** | 1.0012 | 0.9996 |
| Ham | 17.0756 | 17.1056 | **17.0749** | 17.1139 |
| MedicalImages | **3.4197** | 3.6215 | 3.5611 | 3.4369 |
| MiddlePhalanxOutlineAgeGroup | 0.2339 | **0.2338** | 0.2339 | 0.2339 |
| MiddlePhalanxOutlineCorrect | 0.3349 | 0.3348 | 0.3348 | **0.3348** |
| MiddlePhalanxTW | **0.2172** | 0.2174 | 0.2173 | 0.2173 |
| MoteStrain | **18.8197** | 19.0501 | 19.0500 | 19.0499 |
| ProximalPhalanxTW | 0.1585 | **0.1585** | 0.1586 | 0.1586 |
| RefrigerationDevices | 120.0953 | **114.3597** | 118.9731 | 115.0863 |
| ScreenType | 104.6496 | 104.7925 | 105.0297 | **104.1625** |
| ShapeletSim | 121.2170 | 120.8727 | 120.6381 | **120.4151** |
| ShapesAll | **25.5056** | **25.5056** | **25.5056** | **25.5056** |
| SmallKitchenAppliances | 123.8676 | 123.7506 | **123.3957** | 123.5583 |
| SonyAIBORobotSurface1 | 4.6162 | **4.6157** | 4.6566 | 4.6330 |
| SonyAIBORobotSurface2 | 8.8879 | 8.8856 | 8.8813 | **8.8808** |
| SyntheticControl | 8.6781 | 8.7188 | **8.6778** | 8.6984 |
| Trace | 2.9488 | 2.9490 | 2.9518 | **2.9483** |
| TwoLeadECG | 0.8646 | 0.8646 | 0.8646 | **0.8644** |
| Wine | 0.0686 | 0.0682 | 0.0680 | **0.0678** |
| WordSynonyms | **27.5408** | **27.5408** | **27.5408** | **27.5408** |
| Worms | **451.2960** | **451.2960** | **451.2960** | **451.2960** |
| WormsTwoClass | 78.5920 | 78.4135 | 78.7172 | **78.3617** |
| Count | 20 | 19 | 17 | 21 |

THE USE OF LARGE LANGUAGE MODELS

Large Language Models (LLMs) were employed for proofreading and typographical error correction in this study.

