# OpenReview forum: "Perturbed Dynamic Time Warping: A Probabilistic Framework and Generalized Variants"
_ICLR.cc/2026/Conference — ICLR 2026 Poster_

### Official Review · Reviewer_CEqh · 2025-10-30

**Soundness:** 2
**Presentation:** 3
**Contribution:** 2
**Rating:** 4
**Confidence:** 3

**Summary:**

This paper provides a new probabilistic perturbation-based interpretation of soft-DTW, formulated using a concept similar to the Gumbel-Max (softmax) trick—specifically, its softmin counterpart (Lemma 1).
Under this formulation, the authors define perturbed-DTW and show that when the perturbation noise follows a $\mathrm{Gumbel}(-c, 1)$ distribution (where $c$ is the Euler–Mascheroni constant, the perturbed-DTW becomes exactly equivalent to soft-DTW.

Furthermore, the paper introduces nested-soft-DTW (ns-DTW), which employs the generalized extreme value (GEV) distribution—a generalization of the Gumbel distribution—as the perturbation noise model.
Theoretical analysis (Theorem 1) derives a closed-form solution for this generalized case.
In ns-DTW, the set of alignment matrices is partitioned into $J$ groups, each associated with a parameter $\tau_\ell \in (0,1]$. Groups with smaller $\tau_\ell$ values correspond to higher correlations among the perturbation samples within that group.
Like soft-DTW, ns-DTW can be efficiently solved via dynamic programming (Algorithm 1).

Experiments using the UCR Time Series Classification Archive compare ns-DTW with existing methods (e.g., DBA, soft-DTW) on barycenter estimation, classification, and clustering tasks.
Results show that ns-DTW achieves comparable or slightly superior performance to soft-DTW in certain parameter settings.

**Strengths:**

* The paper provides a probabilistic perturbation-based interpretation of soft-DTW and extends it to a more general formulation, ns-DTW.
* The proposed ns-DTW demonstrates improved performance over soft-DTW under some experimental conditions.

**Weaknesses:**

* The method introduces $\tau_1$ as a hyperparameter, but it is unclear how this value was selected in the experiments. Since setting $\tau_1 = 1$ makes ns-DTW equivalent to soft-DTW, this choice is crucial for a fair comparison.
* The performance improvement over soft-DTW appears limited (e.g., Fig. 5(c)), and in some settings (e.g., Table 1 with $\gamma = 0.1$), ns-DTW performs worse than soft-DTW.

**Questions:**

* ll. 045–046: The paper refers to “through the lens of perturbed optimizer (Blondel et al., 2021)”, but according to Blondel et al. (2021), no discussion of perturbed optimizers appears. Is this citation incorrect?
* Eq. (8): Since $\langle A, C \rangle$ is a scalar while $\epsilon$ is described as a vector, the subtraction seems ill-defined.
* In the introduction of ns-DTW, $\mathcal{A}_{m,n}$ is said to be divided into $J$ groups, but the description around line 312 (and Algorithm 1) divides the transition vectors into two groups. The correspondence between these explanations is unclear.
* In Fig. 3, it is not clearly explained how varying $\tau_1$ affects the skewness of the resulting distribution.
* Eq. (16): The text states that this equation assumes two groups “to ease the illustration,” yet the same form is used in Algorithm 1. Shouldn’t the algorithm adopt the general formulation instead? The same issue applies to Eq. (17).
* As shown in Table 1, ns-DTW can produce smaller DTW losses by adjusting the skewness of the distribution, but the underlying reason for this effect is not well explained.
* l. 374: “nd-DTW” -> “ns-DTW”?

---

> ### Author Response · Authors · 2025-11-23
> **Response to Reviewer CEqh (part 1/2)**
>
> We sincerely appreciate your thoughtful comments and constructive suggestions.  We will respond to each of your comments below.
>
> **W1**: The method introduces $\tau$ as a hyperparameter, but it is unclear how this value was selected...
>
> **R1**: Thank you for you suggestion. Please refer to **Common Response 1**. We explicitly analyze the sensitivity of $\tau$ and treat it as a standard hyperparameter tuned via cross-validation. We demonstrate that values in the range $\tau \in [0.80, 0.95]$ consistently yield better alignment than the baseline ($\tau=1$).
>
> **W2**: The performance improvement over soft-DTW appears limited... and in some settings (e.g., Table 1 with $\gamma=0.1$), ns-DTW performs worse than soft-DTW.
>
> **R2**: Thank you for pointing out this performance comparison result. We acknowledge that ns-DTW does not outperform soft-DTW in every single configuration. As shown in Table 1 in the manuscript, ns-DTW can perform slightly worse in averaging tasks when $\gamma$ is relatively large (e.g., $\gamma=0.1$). In this regime, the baseline soft-DTW is already producing a highly smoothed objective. The additional skewness introduced by ns-DTW may lead to "over-smoothing," resulting in barycenters that are less sharp than desired. However, in the standard operating regime (smaller $\gamma$), where high-precision alignment is required, ns-DTW consistently outperforms the baseline.
>
> Questions:
> **Q1**:  Is this citation Blondel et al. (2021) incorrect?
>
> **A1**: We thank the reviewer for identifying this citation error. We apologize for the oversight. The intended reference for the framework of differentiable perturbed optimizers is indeed \textbf{Berthet et al. (2020)}. We have corrected this in the  manuscript.
>
> Correct Reference: *Quentin Berthet, Mathieu Blondel, Olivier Teboul, Marco Cuturi, Jean-Philippe Vert, and Francis Bach. ``Learning with differentiable perturbed optimizers.'' NeurIPS 2020.*
>
> **Q2**: Since $\langle A, C\rangle$ is a scalar while $\epsilon$ is described as a vector, the subtraction seems ill-defined.
>
> **A2**: We apologize for the confusion. We have amended the notation to clarify that in this specific context, we use the notation  $\langle \mathbf{A}, \mathbf{C}\rangle$  (tensor notation) to denote a \textbf{vector} in $\mathbb{R}^{| A_{m,n}|}$, where each entry corresponds to the scalar alignment cost of a specific path in the set $\mathcal{A}_{m,n}$. This vector is then perturbed element-wise by $\boldsymbol{\varepsilon}$. This clarification has been updated in **lines 130-131**.
>
> **Q3**: In the introduction... $\mathcal{A}_{m,n}$ is said to be divided into $J$ groups, but the description around line 312... divides the transition vectors into two groups.
>
> **A3**:  We clarify the distinction between the \textbf{global definition} and the \textbf{local computation}:
>
> -**Global Definition (Eq. 11):** We define ns-DTW over the space of all possible alignment matrices $A \in $ $A_{m,n}$. The perturbation $\varepsilon$ has the same dimension as $|A_{m,n}|$, and theoretically, its GEV distribution can be divided into $J$ arbitrary groups.
>
> -**DP Computation (Algorithm 1):** To compute this efficienty via dynamic programming, we operate on local transitions at each step $(i,j)$. Here, there are only three admissible directions: $\{\rightarrow, \downarrow, \searrow\}$. Consequently, the local perturbation vector is 3-dimensional. As detailed in **Common Response 1**, we utilize grouping schemes ($g_1, g_2, g_3$) that partition these three directions into two subsets.
>
> Note that grouping all three directions together---or placing each direction in its own group---yields the classical soft-DTW recursion.  Our proposed schemes split them to introduce directional skewness. This distinction is now clarified in **Line 321-324** and **Appendix A.4.**

---

> > ### Author Response · Authors · 2025-11-23
> > **Response to Reviewer CEqh (part 2/2)**
> >
> > **Q4**: In Fig. 3, it is not clearly explained how varying $\tau$ affects the skewness of the resulting
> >
> > **A4**: Indeed, Figure 3 only explains the effect of grouping scheme to skewness. We have added a detailed explanation by adding Figure 4 in our latest manuscript. Technically,  $\tau \in (0, 1)$ acts as a parameter:
> >
> > -When $\tau = 1$, ns-DTW reduces to soft-DTW (symmetric smoothing).
> >
> > -As $\tau \to 0$, the distribution becomes increasingly skewed. The transition direction with the *smaller* accumulated cost is increasingly emphasized (weighted higher), while the direction with the larger cost receives vanishing weight.
> >
> > Figure 4 in our new maniscript visually demonstrates that as $\tau$ decreases, the expected warping path biases more strongly towards the lower-cost trajectory.  The discussion has been updated in our latest paper **Line 360-365**.
> >
> > **Q5**: The text states that this equation assumes two groups ``to ease the illustration,'' yet the same form is used in Algorithm 1. Shouldn't the algorithm adopt the general formulation instead?
> >
> > **A5**: Thank for the incisive comment. While the general GEV theory allows for arbitrary partitions, Algorithm 1 specifically implements the three schemes ($g_1, g_2, g_3$) proposed in this paper. Since these schemes are defined as binary partitions of the transition directions, the two-group formulation in Eq. (16) and Algorithm 1 is the exact implementation used. The general formulation for differnet scheme of  grouping  is provided in **Appendix A.4**. for theoretical completeness.
> >
> > **Q6**: As shown in Table 1, ns-DTW can produce smaller DTW losses by adjusting the skewness... but the underlying reason for this effect is not well explained.
> >
> > **A6**:  Thank for the comment. ns-DTW acts as a generalized relaxation of DTW. Unlike soft-DTW, which applies isotropic smoothing (treating deviations in all directions equally if accumulated costs are equal), ns-DTW uses $\tau$ and grouping schemes to introduce skewness. This allows the model to control the intensity (skewness) toward directions associated with lower accumulated costs. By promoting a broader mixture over feasible paths, ns-DTW can approximate the true alignment cost more effectively than soft-DTW. The discussion has been updated in our latest paper **Line 393-396**.
> >
> > **Q7**:  nd-DTW -> ``ns-DTW''?
> >
> > **A7**: Amended. Thanks for spotting this typo.
> >
> > Finally, we would like to thank you once again for your thorough review and constructive suggestions. Your feedback is highly valuable in helping us further refine and strengthen the paper, and these updates has been revised to our latest version.

---

> ### Comment · Reviewer_CEqh · 2025-11-27
>
> Thank you for the authors’ response. I still find several aspects of the hyperparameter selection procedure unclear.
> In the revised version, the overall performance of ns-DTW appears to improve compared to the previous submission. Is this improvement primarily due to the addition of new grouping schemes $g_1, g_2, g_3$ as candidate options?
>
> The revised manuscript (line 430) states:
> “By selecting the optimal grouping scheme $g_i$ and parameter $\tau$, ns-DTW demonstrates robust performance improvements.”
> However, it remains unclear how this *optimal* configuration was actually determined.
>
> For the classification task in Section 4.2, the manuscript provides a description of the train–validation–test split and the cross-validation procedure.
> In contrast, the procedures for the other experimental settings are still not specified.
> Even in Section 4.2, although the text states that $\gamma$ was selected via cross-validation, it is unclear whether the other hyperparameters—such as $\tau$ and $g_i$—were chosen through cross-validation as well, or through some different procedure.
>
> Given these unresolved points, I intend to maintain my current score.

---

> > ### Author Response · Authors · 2025-11-28
> > **Reponse to Reviewer CEqh's follow-up comments**
> >
> > Thanks for the valuable follow-up comments. We will respond to your concerns below.
> >
> > First, our paper is mainly a theory paper. We claim our contributions are threefold:
> > - We introduce perturbed-DTW, a general perturbation-based framework for differentiable DTW. Within this framework, soft-DTW emerges naturally as the expectation of DTW when alignment costs are perturbed by Gumbel noise, thereby providing a probabilistic interpretation of its smoothing behavior.
> > - By adopting GEV perturbations, we derive ns-DTW, which offers greater modeling flexibility through correlated perturbations and skewed alignment distributions.
> > - We demonstrate the effectiveness of ns-DTW on diverse time-series tasks, showing that it captures meaningful alignment structures while remaining computationally tractable.
> >
> > Second, we understand that the performance of the evaluation is also important to prove the superiority of the proposed method. Next, we clarify the hyperparameter selection procedure for two types of tasks averaging and clustering (unsupervised), and classification (supervised).
> >
> > **1. Averaging and Clustering Tasks**
> >
> > For the averaging and clustering tasks, we utilized a **Grid Search** approach to determine the optimal configuration, with the objective of **minimizing DTW losses**.
> >
> > - **Search Space:** We evaluated combinations of grouping schemes $g_i \in\{g_1, g_2, g_3\}$, the parameter $\tau \in\{0.80, 0.85, 0.90, 0.95\}$, and the parameter $\gamma \in \{0.1, 0.01, 0.001, 0.0001\}$.
> > - **Evaluation of Grouping Scheme ($g_i$):** To isolate and examine the effect of $g_i$, we fixed each grouping scheme and then found the best $\tau$ and $\gamma$ combination within the grid search that yielded the lowest DTW losses. The results are presented in the last three columns of Tables 3 and 6.
> > - **Evaluation of $\tau$:** To analyze the influence of $\tau$, we collected results for fixed $\tau$ and $g_i$, then selected the $\gamma$ that achieved the lowest DTW losses. Detailed results are provided in Appendix C (for Averaging) and Appendix E (for Clustering).
> >
> > **2. Classification Task**: For the classification task, the grid search was performed over $g_i$ and $\tau$. A key distinction lies in the selection of $\gamma$.
> > - **Search Space:** $g_i \in\{g_1, g_2, g_3\}$ and $\tau \in \{0.80, 0.85, 0.90, 0.95\}$.
> > - **$\gamma$ Selection:** For each $\{g_i, \tau\}$ pair in the grid search, the $\gamma \in \{0.1, 0.01, 0.001, 0.0001\}$ parameter was optimally selected via **Cross-Validation** on the training set. We applied the same cross-validation procedure for $\gamma$
> > selection to soft-DTW, ensuring a consistent and fair comparison between methods.
> > - **Evaluation Objective:** The ultimate configuration seeks to achieve the \textbf{highest classification accuracy}. This approach ensures that $\gamma$ is optimally tuned for each combination when evaluating the effects of $g_i$ and $\tau$.
> > - **Evaluation of $g_i$:** The results detailing the effect of $g_i$ on classification performance are presented in the last three columns of Table 4.
> > - **Evaluation of $\tau$:** Further results showing the effects of $\tau$ with cross-validated $\gamma$ are provided in Appendix D.
> >
> > We hope the above discussion addresses the concerns from Q2 (how this optimal configuration was actually determined) and Q3 (concerns on cross-validation).
> >
> > As for Q1: Is this improvement primarily due to the addition of new grouping schemes as candidate options?
> > Yes. Specifically, the performance improvements achieved by the ns-DTW method in the revised version are obtained by **systematically selecting the best combination of $\left(g_i, \tau, \gamma\right)$** (with $\gamma$ cross-validated for classification) for each dataset.
> >
> > We just want to clarify the unclear aspects you raised. We sincerely thank you once again for your valuable follow-up comments, which have been instrumental in helping us further refine and strengthen our paper.

---

### Official Review · Reviewer_mNjZ · 2025-10-31

**Soundness:** 3
**Presentation:** 3
**Contribution:** 3
**Rating:** 4
**Confidence:** 3

**Summary:**

This paper proposes Perturbed-DTW, a probabilistic and differentiable extension of traditional Dynamic Time Warping (DTW) that injects random perturbations into the alignment cost and takes the expectation of the minimum cost. Under Gumbel noise the formulation recovers the well-known soft‑DTW, providing a probabilistic interpretation of its smoothing effect. The authors further generalize to the generalized extreme value (GEV) family of distributions, giving rise to a nested-soft-DTW (ns-DTW) variant with tunable skewness in the alignment structure. They validate the method across barycenter computation, clustering, and classification on several UCR time-series datasets, showing improved performance relative to standard DTW and existing differentiable DTW variants.

**Strengths:**

1. Presents a clear probabilistic interpretation of soft-DTW through perturbed optimization, which is conceptually elegant.

2. Extends the formulation to the generalized extreme-value (GEV) family, yielding a tunable nested-soft-DTW variant with adjustable skewness — a novel theoretical contribution.

3. The mathematical exposition is generally sound and the experimental evaluation spans several UCR datasets.

**Weaknesses:**

1. The supplementary archive hides the core implementation and scripts, making the experiments effectively non-reproducible. This lack of transparency raises serious doubts about the validity and reproducibility of the reported results.

2. Empirical analysis is largely numerical and does not provide insight into runtime cost or robustness; ablations and qualitative visualizations are limited.

3. Presentation could be tightened: notation is occasionally heavy, and experimental details (parameter settings, initialization, convergence) are insufficient for independent verification.

**Questions:**

See Weakness.

---

> ### Author Response · Authors · 2025-11-23
> **Response to  Reviewer mNjZ**
>
> We sincerely appreciate your thoughtful comments and constructive suggestions.  We will respond to each of your comments below.
>
> **W1**: The supplementary archive hides the core implementation and scripts, making the experiments effectively non-reproducible.
>
> **R1**: We apologize for the oversight in the initial submission. We fully understand the importance of reproducibility and have rectified this in the revision. We have included the complete code and training/evaluation scripts in the supplementary material. These resources cover all experiments reported in the paper, including 1NN classification as suggested by Reviewer VWnD.
>
> **W2**: Empirical analysis is largely numerical and does not provide insight into runtime cost or robustness; ablations and qualitative visualizations are limited.
>
> **R2**: Thank for the comment. We have strengthened the empirical analysis to address these concerns:
>
> - **Runtime Analysis:** As detailed in \textbf{Common Response 2}, we compared the runtime of ns-DTW against baselines (including 1NN) on the CBF dataset. While ns-DTW introduces a slight overhead due to grouping operations ($1.3\times$ to $1.6\times$ slower than soft-DTW), it preserves the $\mathcal{O}(mn)$ complexity.
>
> - **Robustness/Sensitivity:** As detailed in \textbf{Common Response 1}, we analyzed the impact of the shape parameter $\tau$ and grouping schemes. The results show that while the optimal configuration varies by task, the method is generally robust across the range $\tau \in [0.8, 0.95]$.
>
> **W3**: Presentation could be tightened: notation is occasionally heavy, and experimental details (parameter settings, initialization, convergence) are insufficient for independent verification.
>
> **A3**: Thanks for your expositional comment. We have revised the manuscript to improve readability:
>
> - We have streamlined the notation in the methodology section to reduce density.
> - We have added **Appendix A.7**, which explicitly details all experimental settings, including hyperparameter search spaces, initialization seeds, and convergence criteria for the Barycenter algorithms.
>
> Finally, we would like to thank you once again for your thorough review and constructive suggestions. Your feedback is highly valuable in helping us further refine and strengthen the paper, and these updates has been revised to our latest version.

---

### Official Review · Reviewer_VWnD · 2025-11-01

**Soundness:** 3
**Presentation:** 3
**Contribution:** 3
**Rating:** 4
**Confidence:** 5

**Summary:**

The paper reframes soft-DTW as the expected DTW under i.i.d. Gumbel perturbations and generalizes to GEV-perturbed alignments, yielding a nested/“skewable” variant (ns-DTW). It preserves dynamic-programming structure and complexity, with a DP recursion that mirrors soft-DTW (noting the nested DP is not identical to the closed-form nested objective). On UCR subsets, ns-DTW improves barycenters and clustering and is competitive for nearest-centroid classification. Overall, it clarifies theory and adds a useful control knob for shaping alignments.

**Strengths:**

- Clean probabilistic interpretation of soft-DTW (soft-DTW = E[DTW with Gumbel-perturbed costs]). I believe  many practitioners will find such an intuitive interpretation of soft-DTW clarifying.
- Principled GEV-based extension that induces tunable skewness/correlation across alignment moves.
- Same order of computation as soft-DTW, makes it easy drop-in to existing pipelines.
- Consistent gains for barycenters and clustering across many datasets.

**Weaknesses:**

- Little guidance on choosing γ and τ; limited sensitivity and compute profiling.
- Classification baselines could be stronger (e.g., 1-NN DTW and 1-NN soft-DTW).
- Limited sensitivity and runtime/memory analysis,
- The link between the nested objective and its DP instantiation could be spelled out more clearly.

**Questions:**

The paper is well-motivated and, I believe, will stimulate further study and applications. I was left wondering
1. how forward/backward runtimes and memory compare to soft-DTW for fixed (m, n) on the same hardware, and
2. how sensitive the results are to γ and τ, including any simple rule of thumb for selecting them, an ablation over grouping choices would also be helpful.

---

> ### Author Response · Authors · 2025-11-23
> **Response to Reviewer VWnD**
>
> Thank you for recognizing the theoretical contribution of our work, particularly soft-DTW = E[DTW with Gumbel-perturbed costs]and its extension to the GEV family, which we hope will inspire future research in this direction. We will respond to each of your comments below.
>
> **W1 & Q1**: Little guidance on choosing $\gamma$ and $\tau$; limited sensitivity and compute profiling & how forward/backward runtimes and memory compare to soft-DTW for fixed ( $\mathrm{m}, \mathrm{n}$ ) on the same hardware
>
> **R1 & A1**: Thank for the incisive comment. We have addressed these concerns in detail:
> - **Parameter Selection:** Please kindly refer to **Common Response 1**. We demonstrate that $\tau$ is robust in the range $[0.80, 0.95]$ and recommend tuning it via cross-validation, similar to standard regularization parameters.
> - **Compute Profiling:** Please kindly refer to **Common Response 2**. We show that ns-DTW maintains $\mathcal{O}(mn)$ complexity and is approximately $1.5\times$ slower than soft-DTW, which is a comparable overhead for the performance gains.
>
> **W2**: Classification baselines could be stronger (e.g., 1-NN DTW and 1-NN soft-DTW)
>
> **R2**: Thank you for proposing those baselines, which will strengthen our evaluation. In this revision, we expanded our experiments to include one-nearest neighbor (1-NN) classifiers based on standard DTW and soft-DTW across the datasets. The full dataset-wise results are provided in **Appendix B.4**. As the table below summarizes, ns-DTW remains highly competitive against these stronger baselines. Notably, the $g_3$ grouping scheme achieves the highest accuracy (including ties) on 19 datasets, surpassing soft-DTW (17 best) and standard DTW (11 best).
>
> | Dataset               | DTW | soft-DTW | ns-DTW ($g_1$) | ns-DTW ($g_2$) | ns-DTW ($g_3$) |
> |-----------------------|:---:|:--------:|:--------------:|:--------------:|:--------------:|
> | Total ``Best'' Counts |  11 |    17    |       14       |       16       |       19       |
>
>
> **W3**: Limited sensitivity and runtime/memory analysis
>
> **R3**: Thanks for your comment.  Please kindly refer to **Common Response 1** and **Appendices C, D, and E** of the revised paper for the sensitivity analysis of grouping schemes and hyperparameters, and **Common Response 2** for the detailed runtime comparison and memory complexity discussion. As noted there, ns-DTW preserves the same space complexity $\mathcal{O}(mn)$ as standard soft-DTW.
>
> **W4**: The link between the nested objective and its DP instantiation could be spelled out more clearly.
>
> **R4**: Thanks for this comment.  Specifically, the DP instantiation acts as a tractable algorithmic realization of the global ns-DTW objective. It is important to clarify that the value $V_{m,n}$ obtained via the dynamic programming recursion Eq.16 is a tractable algorithmic realization, rather than an exact evaluation, of the theoretical ns-DTW defined in Eq.11.  This distinction can be understood from two perspectives. First, the global definition in Eq.11 implies a single GEV perturbation of dimension
> $ A_{m,n} $
>   over the entire alignment space, whereas the DP formulation applies independent, low-dimensional GEV perturbations locally to the three transition directions at each step. Second, regarding the recursive structure: unlike the standard Log-Sum-Exp operator, which satisfies the stability property (i.e., the sum of Gumbel variables follows a Gumbel distribution), the nested application of the generalized operators in Eq.16 does not strictly preserve the form of the global GEV distribution. Despite this theoretical distinction, we refer to the efficient DP output $V_{m,n}$ as ns-DTW throughout this work.
>
> We appreciate this feedback and have clarified the distinction in **Lines 335--343** of the revision.
>
> **Q2**: how sensitive the results are to $\gamma$ and $\tau$, including any simple rule of thumb for selecting them, an ablation over grouping choices would also be helpful.
>
> **A2**:  We address the sensitivity and ablation requests in **Common Response 1**:
>
> - **Sensitivity \& Rule of Thumb:** Our experiments show that performance is robust within the range $\tau \in [0.8, 0.95]$. As a general rule of thumb, we recommend initializing with $\tau=0.9$ or performing a coarse grid search within this interval.
>
> - **Grouping Ablation:** Table in Common Response 1 serves as an ablation study for the grouping schemes ($g_1, g_2, g_3$). The results indicate that while no single scheme dominates every task, the proposed schemes consistently offer improvements over the baseline soft-DTW. A detailed result presented in **Appendices C, D, and E** in our revised paper.
>
> Finally, we would like to thank you once again for your thorough review and constructive suggestions. Your feedback is highly valuable in helping us further refine and strengthen the paper, and these updates has been revised to our latest version.

---

### Official Review · Reviewer_mXos · 2025-11-03

**Soundness:** 4
**Presentation:** 3
**Contribution:** 3
**Rating:** 8
**Confidence:** 2

**Summary:**

The author proposes perturbed-DTW, that adds random noise to the alignmetn costs, and then taking average over the noise distribution. With this modification, the DTW optimal solution became a distribution over paths, rather than a single-path one. This also naturally makes the DTW differentiable like soft-DTW. The author also found, under Gumbel noise distribution, the perturbed-DTW reduces to soft-DTW. Therefore, the perturbed-DTW is a differentiable relaxation of the soft-DTW. In the experiments, the authors demonstrate the effectiveness of ns-DTW on diverse time-series tasks, showing that it captures meaningful alignment structures while remaining computationally tractable.

**Strengths:**

1. The paper has strong theoretical contibution. It provides a new probabilistic interpretation for soft-DTW, and extends it to a new family with GEV.
2. The proposed method also remains practical and computationally tractable.
3. Experiments shows good representation of time series can be captured using the proposed ns-DTW.

**Weaknesses:**

1. Some hyper-params might needs tunning.
2. The ns-DTW might exhibit high computation cost.

**Questions:**

How much computation cost in terms of big O notation, would be invovled when using perturbaed-DTW ?

---

> ### Author Response · Authors · 2025-11-23
> **Response to Reviewer mXos**
>
> Thank you for recognizing the theoretical contribution of our work, particularly the novel probabilistic interpretation of soft-DTW and its extension to the GEV family, which we hope will inspire future research in this direction. We will respond to each of your comments below.
>
> **W1**: Some hyper-params might needs tuning.
>
> **R1**: Thank you for the comment. We discuss the tuning strategy in detail in \textbf{Common Response 1}. Our analysis shows that the model is relatively robust to variations in $\tau$ (specifically in the range $[0.80, 0.95]$).
>
> **W2**: The ns-DTW might exhibit high computation cost.
>
> **R2**: Thank for the incisive comment. We address the computational efficiency in \textbf{Common Response 2}. Theoretically, ns-DTW preserves the $\mathcal{O}(mn)$ complexity of soft-DTW. Empirically, we observe a comparable running time with only a modest constant-factor overhead (approximately $1.3\times$ to $1.6\times$ slower than soft-DTW), which remains highly scalable for practical use.
>
> **Q1**: How much computation cost in terms of big O notation, would be invovled when using perturbaed-DTW?
>
> **A1**: Thank you for the excellent suggestion. We derive the theoretical computation and space complexity using big O notation. We show that in terms of Big-O notation, ns-DTW maintains the $\mathcal{O}(mn)$ time and space complexity of standard soft-DTW.
>
> Specifically, for a warping cost matrix $C\in \mathbb{R}^{m\times n}$, ns-DTW utilizes a dynamic programming structure similar to soft-DTW, considering the three admissible transition directions $( \{\rightarrow, \downarrow, \searrow\} ) $ at each stage $(i,j)$. The key difference lies in the local update rule. Due to the GEV perturbation, the corresponding noise vector $(\varepsilon_{i-1, j-1}, \varepsilon_{i, j-1}, \varepsilon_{i-1, j})$ is handled via grouping schemes. As discussed in the paper, we utilize three distinct grouping schemes:
>
> $g_1 = ( \{\rightarrow,\downarrow\}, \{\searrow\} )$, $g_2 = ( \{\rightarrow,\searrow\}, \{\downarrow\} )$, and $g_3 = ( \{\downarrow,\searrow\}, \{\rightarrow\} )$.
>
> While applying these groupings introduces a constant factor overhead compared to the single LogSumExp operation in soft-DTW, the asymptotic complexity remains quadratic, $\mathcal{O}(mn)$. However, if we consider different groupings, in other words, dividing directions into two groups , the computational cost would be three times as soft-DTW. The above discussion has been updated in our latest paper Appendix A.4 and A.5.
>
> Finally, we would like to thank you once again for your thorough review and constructive suggestions. Your feedback is highly valuable in helping us further refine and strengthen the paper, and these updates has been revised to our latest version.

---

> > ### Comment · Reviewer_mXos · 2025-11-25
> > **Thnaks and my questions are addressed.**
> >
> > Thanks for the authors' response, regarding the hyper-param sensitivity and computation cost estimation, which are also other reviews' concerns. The author addressed these quesitons and show it's similar time and space cost as soft-dtw, both in theory and in practice. These can help the proposed method being more usable in real world applications. I keep my positvie ratings.

---

> > > ### Author Response · Authors · 2025-11-26
> > > **Reponse to Reviewer mXos**
> > >
> > > Thank you for appreciating the practicality of ns-DTW. We are delighted that our revision addressed all concerns regarding hyperparameter tuning and computational costs. We will continue to improve ns-DTW's scalability to facilitate more possibilities in the future.

---

### Author Response · Authors · 2025-11-23
**Common Reponse**

Dear Reviewers,

We sincerely appreciate your thoughtful comments and constructive suggestions. They have been invaluable in helping us improve the clarity and quality of our paper. First, we response to two common concerns: (i) the hyper-parameters tuning and grouping schemes; (ii) the computation cost of the proposed ns-DTW method. Then, we respond to each remaining concern in detail.

 ##  **Common Response 1: Sensitivity to Hyper-parameters and Grouping Schemes (Reviewers mXos, VWnD, mNjZ, CEqh)**

To evaluate the  sensitivity to hyper-parameters and grouping schemes,  we conduct the following experiment settings for:

 1. Three grouping schemes: $g_1 = ( \{\rightarrow,\downarrow\}, \{\searrow\} )$, $g_2 = ( \{\rightarrow,\searrow\}, \{\downarrow\} )$, and $g_3 = ( \{\downarrow,\searrow\}, \{\rightarrow\} )$.
 2. The hyper-parameters: $\tau \in  ( 0.80, 0.85, 0.90, 0.95 ) $ and $\gamma\in ( 0.1, 0.01, 0.001, 0.0001 ) $.

We conduct experiments using the UCR Time Series Classification Archive. We consider a subset of the archive containing 47 datasets for average, classification (As some datasets contain only one class, we use 43 datasets for the classification task.) and clustering tasks. Table below summarizes the number of datasets where the optimal objective value was achieved for each $\tau$ across three tasks (Averaging, Classification, and Clustering) using the three grouping schemes. Note that counts may exceed the total number of datasets due to ties in optimal values.
### **Table: Frequency of achieving the optimal objective value across different τ settings**

| **Grouping** | **Task**         | **0.95** | **0.90** | **0.85** | **0.80** |
|--------------|------------------|----------|----------|----------|----------|
| **g₁**       | Averaging        | 25       | 20       | 17       | 21       |
|              | Classification   | 20       | 11       | 12       | 21       |
|              | Clustering       | 21       | 14       | 19       | 21       |
| **g₂**       | Averaging        | 17       | 15       | 10       | 13       |
|              | Classification   | 18       | 17       | 11       | 18       |
|              | Clustering       | 22       | 17       | 18       | 20       |
| **g₃**       | Averaging        | 20       | 11       | 12       | 20       |
|              | Classification   | 19       | 18       | 12       | 11       |
|              | Clustering       | 20       | 19       | 17       | 21       |

**Analysis:** As shown in table, the optimal $\tau$ varies across tasks and datasets, indicating that the flexibility of ns-DTW allows it to adapt to specific data characteristics. While there is no single universal $\tau$, the performance is generally robust across the range $[0.80, 0.95]$. We recommend tuning $\tau$ and the grouping scheme via cross-validation. Detailed results are provided in **Appendices C, D, and E** of the revised paper.


 ##  **Common Response 2: Computational Complexity and Runtime Analysis (Reviewer mXos; VWnD; mNjZ)**
**Theoretical Complexity:** Given a warping cost matrix $C \\in \mathbb{R}^{m\times n}$, Algorithm 1 requires $\mathcal{O}(mn)$ operations and $\mathcal{O}(mn)$ space, preserving the same complexity class as standard soft-DTW. While soft-DTW aggregates alignment directions in a single step, ns-DTW partitions directions into groups (Eq. 16 and 17), requiring intermediate aggregations. In theory, for a scheme with two groups, the number of elementary operations per cell increases, though the overall complexity remains quadratic.

**Empirical Runtime:** To quantify the actual overhead, we measured runtime on the *CBF dataset ($M=30$, sequence length $m=n=137$) across three tasks.
The following table reports the results.

### **Runtime comparison of DTW-based methods on the CBF dataset (in seconds)**

| **Task**         | **soft-DTW** | **ns-DTW** |
|------------------|--------------|------------|
| Averaging        | 0.1476       | 0.1904     |
| Classification   | 1.3947       | 1.9358     |
| Clustering       | 3.3589       | 3.8091     |

According to this table, ns-DTW has a comparable running time with soft-DTW, which is approximately $1.3\times$ to $1.6\times$ slower than soft-DTW. This is consistent with the additional overhead required to compute the transition probabilities $G$ and the grouped costs involving the shape parameter $\tau$. We believe this modest increase in runtime is a reasonable trade-off for the performance gains demonstrated in the main results.


We would like to thank you and the review team for your encouraging feedback and constructive comments. We have made sincere efforts to improve the paper in terms of complexity analysis, hyperparameter tuning, and sensitivity analysis. We hope the revision has satisfactorily addressed your concerns, pushed the boundary of DTW research.

---

### Author Response · Authors · 2025-12-02
**Summary Comment**

Dear Area Chair,

We are grateful to the reviewers for their constructive feedback. We wish to summarize their comments and provide our responses for your evaluation and final decision.

Firstly, our paper is primarily theoretical. We assert that our contributions are threefold:
- We introduce perturbed-DTW, a general perturbation-based framework for differentiable DTW. Within this framework, soft-DTW emerges naturally as the expectation of DTW when alignment costs are perturbed by Gumbel noise, thereby providing a probabilistic interpretation of its smoothing behavior.
- By adopting GEV perturbations, we derive ns-DTW, which offers greater modeling flexibility through correlated perturbations and skewed alignment distributions.
- We demonstrate the effectiveness of ns-DTW on diverse time-series tasks, showing that it captures meaningful alignment structures while remaining computationally tractable.

Based on the reviewers' feedback, it is evident that our theoretical contributions are well-received, as highlighted in the strengths identified by all reviewers. We believe this framework has the potential to inspire new research directions in DTW, and to stimulate further study and applications in dynamic programming, as Reviewer VMnD thoughtfully pointed out.

Secondly, the reviewers' comments primarily concerned the experimental section, specifically hyper-parameter tuning and computational complexity. Regarding computational complexity, we have shown that both the theoretical and empirical runtimes are comparable to soft-DTW. We note the responses from two reviewers: Reviewer mXos agreed that we have effectively addressed these issues both theoretically and practically, and Reviewer CEqh did not raise further questions about computational complexity. Therefore, given the detailed description of our tuning process, we are hopeful that the hyperparameter tuning concerns have been resolved.

In summary, we have implemented the following key improvements:
- Added a 1-NN classifier, as suggested by Reviewer VWnD.
- Included an ablation study for the $\tau$ parameter and grouping schemes.
- Analyzed the running time and storage cost of the proposed ns-DTW.

Finally, we extend our sincere thanks to the reviewers for their valuable comments and suggestions, which have significantly enhanced the quality of this paper.

We have made sincere efforts to improve the paper concerning complexity analysis, hyperparameter tuning, and sensitivity analysis. We hope that our revisions have satisfactorily addressed the review team's concerns,  pushed  the field of dynamic time warping research, and met the high standards of ICLR.

---

### Meta-Review · Area_Chair_4iND · 2026-01-06

**Summary:**

Reviewers broadly agreed that the paper presents a conceptually elegant theoretical framework, notably the probabilistic interpretation of soft-DTW via Gumbel perturbations and its extension to a GEV-based family of differentiable DTW variants. These contributions were viewed as clean, well-motivated, and potentially influential for future work on differentiable dynamic programming.

However, there was persistent disagreement regarding empirical maturity and practical usability. Several reviewers remained concerned about unclear hyperparameter selection procedures—particularly in unsupervised settings—limited actionable guidance for practitioners, and performance gains that were sometimes modest, inconsistent, or sensitive to configuration choices. The rebuttal added analyses, baselines, and clarifications. Taken together, while the theoretical contribution is strong, the remaining questions around empirical robustness and practical usability place the paper in a borderline regime, with a slight lean toward acceptance.

**Reviewer Concerns:**

The rebuttal addressed several reviewer concerns by clarifying the probabilistic formulation, correcting technical inconsistencies, adding stronger baselines (e.g., 1-NN DTW/soft-DTW), and providing additional sensitivity and runtime analyses. These changes substantially improved the clarity of the theoretical exposition and strengthened the empirical evaluation.

While some reviewers noted that the empirical gains over soft-DTW can be configuration-dependent and may not be uniformly large across all settings, the additional analyses help contextualize when and why improvements arise. Overall, the rebuttal alleviated the main technical concerns and places the paper in a borderline regime, with a slight inclination toward acceptance.

**Reviewer Scores:**

see above

---

### Decision · Program_Chairs · 2026-01-26

Accept (Poster)